# Single-cell ATAC and RNA sequencing reveal pre-existing and persistent cells associated with prostate cancer relapse

S. Taavitsainen[1], N. Engedal[2], S. Cao[3], F. Handle[4,5], A. Erickson[6], S. Prekovic[7], D. Wetterskog[8], T. Tolonen[1,9], E. M. Vuorinen[1], A. Kiviaho[1], R. Nätkin[1], T. Häkkinen[1], W. Devlies[4,10], S. Henttinen[1], R. Kaarijärvi[11], M. Lahnalampi[11], H. Kaljunen[11], K. Nowakowska[8], H. Syvälä[1], M. Bläuer[1], P. Cremaschi[8], F. Claessens[4], T. Visakorpi[12,13], T. L. J. Tammela[1], T. Murtola[1], K. J. Granberg[1], A. D. Lamb[6,14], K. Ketola[11], I. G. Mills[6,15,16], G. Attard[8], W. Wang[3], M. Nykter[1✉] & A. Urbanucci[2✉]

Prostate cancer is heterogeneous and patients would benefit from methods that stratify those who are likely to respond to systemic therapy. Here, we employ single-cell assays for transposase-accessible chromatin (ATAC) and RNA sequencing in models of early treatment response and resistance to enzalutamide. In doing so, we identify pre-existing and treatment-persistent cell subpopulations that possess regenerative potential when subjected to treatment. We find distinct chromatin landscapes associated with enzalutamide treatment and resistance that are linked to alternative transcriptional programs. Transcriptional profiles characteristic of persistent cells are able to stratify the treatment response of patients. Ultimately, we show that defining changes in chromatin and gene expression in single-cell populations from pre-clinical models can reveal as yet unrecognized molecular predictors of treatment response. This suggests that the application of single-cell methods with high analytical resolution in pre-clinical models may powerfully inform clinical decision-making.

[1] Prostate Cancer Research Center, Faculty of Medicine and Health Technology, Tampere University and Tays Cancer Center, Tampere, Finland. [2] Department of Tumor Biology, Institute for Cancer Research, Oslo University Hospital, Oslo, Norway. [3] Department of Bioinformatics and Computational Biology, The University of Texas MD Anderson Cancer Center, Houston, TX, USA. [4] Molecular Endocrinology Laboratory, Department of Cellular and Molecular Medicine, KU Leuven, Leuven, Belgium. [5] Department of Urology, Division of Experimental Urology, Medical University of Innsbruck, Innsbruck, Austria. [6] Nuffield Department of Surgical Sciences, University of Oxford, Oxford, UK. [7] Division of Oncogenomics, Oncode Institute, The Netherlands Cancer Institute, Amsterdam, The Netherlands. [8] University College London Cancer Institute, London, UK. [9] Department of Pathology, Fimlab Laboratories, Tampere University Hospital, Tampere, Finland. [10] Department of Urology, UZ Leuven, Leuven, Belgium. [11] Institute of Biomedicine, University of Eastern Finland, Kuopio, Finland. [12] Faculty of Medicine and Health Technology, Tampere University and Tays Cancer Center, Tampere, Finland. [13] Fimlab Laboratories, Ltd, Tampere University Hospital, Tampere, Finland. [14] Department of Urology, Churchill Hospital Cancer Centre, Oxford, UK. [15] Patrick G Johnston Centre for Cancer Research, Queen's University of Belfast, Belfast, UK. [16] Centre for Cancer Biomarkers (CCBIO), University of Bergen, Bergen, Norway. ✉email: matti.nykter@tuni.fi; alfonsourbanucci@gmail.com

Prostate cancer (PC) relies on androgen-receptor (AR) signaling for development and progression. Progression on androgen-deprivation therapy (ADT) or AR signaling inhibitors (ARSIs), such as the second-generation AR antagonist enzalutamide (ENZ), leads to castration-resistant (CRPC) or treatment-induced neuroendocrine prostate cancer (NEPC)[1]. The most frequently characterized mechanisms of PC or CRPC resistance to ARSIs, ADT, or both revolve around the restoration of AR signaling via AR overexpression or AR mutations[2–5].

PC is profoundly heterogeneous[6–9] and patients would benefit from methods that differentiate between clinically mild and more aggressive forms of the disease. Although evidence of clonal expansion has been shown[6], most studies to date have characterized genetic mutations[10–12] that do not make allowance for tissue complexity or the biological basis for the emergence of treatment resistance. Furthermore, nongenetic changes in transcriptomics, chromatin structure, and DNA accessibility of transcription factor (TF)-binding motifs are more frequent but less understood in PC drug resistance[2–5]. DNA accessibility is the first layer of gene regulation and transcriptomic changes are now being used to identify molecular predictors of cancer-treatment response[13]. However, most RNA sequencing data are obtained from the bulk of the tumors and therefore cannot account for PC heterogeneity. This is because the transcriptome is the result of several biological processes contributing to differential gene regulation and such processes are not necessarily synchronized in all cells within the tumor bulk[14,15]. The development of single-cell sequencing technologies has enabled a more detailed examination of genomic features in treatment-resistant cancers, but the associated analytical methods are just beginning to reveal their potential.

In this work, we analyze the emergence of resistance in the epithelial-derived component of prostate tumors in ENZ-exposed and -resistant PC cell lines at a single-cell level to explore how heterogeneous PCs respond to ARSIs. Through enrichment analysis of transcriptional signals from molecular gene classifiers derived in this study, we show evidence of treatment-persistent and pre-existing PC cells that can predict treatment response in both primary and advanced patients.

## Results

### Chromatin reprogramming underpins enzalutamide resistance.
To study the molecular consequences of AR signaling suppression and drug-resistance dynamics in PC, we utilized LNCaP parental cell lines, LNCaP-derived ENZ-resistant cell lines RES-A and RES-B generated via long-term exposure to AR-targeting agents[16] (see "**Methods**"), and other independently generated LNCaP- and VCaP-derived models (Fig. 1a, Supplementary Table 1). We hypothesized that chromatin structure would undergo reshaping in ENZ-resistant cells and lead to modification of the transcriptome[17,18].

To determine the contribution of chromatin structure to ENZ resistance, we performed single-cell (sc) assays for transposase-accessible chromatin and sequencing (scATAC-seq) on four samples: (1) LNCaP parental cells (LNCaP), (2) LNCaP exposed to short-term (48 h) ENZ (10 µM) treatment (LNCaP–ENZ48), (3) RES-A, and (4) RES-B (Fig. 1a). We first analyzed the scATAC-seq data as it would have been sequenced in bulk cells (see "**Methods**"). The ATAC-seq signal at transcription-start sites (TSS) decreased in ENZ-resistant cells compared with the parental (average enrichment score 6.3 in resistant cells vs 7.8 in parental cells, $p < 2.2e-16$, Wilcoxon rank-sum test) (Fig. 1b). This pattern was also observed for housekeeping genes, genes from the androgen-response pathway, and genes involved in MYC signaling, suggesting that this pattern is not restricted to a

particular gene set. We noted chromatin opening outside of the TSS region in both resistant cell lines as the potential explanation for the decreased TSS enrichment (Supplementary Fig. 1a). RES-A and RES-B cells shared a large proportion of ENZ-resistance-specific open-chromatin regions not found in parental LNCaP (14% in RES-A and 17% in RES-B). Additionally, RES-A cells had a higher proportion of unique open sites compared with RES-B (19% vs 5%, $p < 2.2e-16$, chi-square test) and LNCaP (19% vs 7%, $p < 2.2e-16$, chi-square test) (Fig. 1c).

We corroborated the extent of chromatin opening and reprogramming in ENZ-resistant cells by performing formaldehyde-assisted isolation of regulatory element (FAIRE) sequencing[19] on the parental LNCaP and RES-A cells subjected to androgen starvation or exposed to androgens, ENZ, or both agents (Supplementary Fig. 1b–g) (see "**Methods**"). Even in this bulk assay, ENZ and androgen starvation appeared to be more significant drivers of reprogramming in RES-A than in parental LNCaP. While there was no significant difference in the total number of open chromatin sites, ENZ-resistant samples had a higher proportion of unique open sites compared with the parental in the presence of androgens (24% vs 12%, $p < 2.2e-16$, chi-square test) (Supplementary Fig. 1d) and in androgen-deprived (castrate) conditions (27% vs 9%, $p < 2.2e-16$, chi-square test) (Supplementary Fig. 1f). Read-distribution analysis (see "**Methods**") demonstrated that the chromatin of ENZ-resistant cells is more open in the presence of androgens ($p < 0.001$, $t$-test) (Supplementary Fig. 1e) and in castrate conditions ($p = 0.022$, $t$-test) (Supplementary Fig. 1g).

Next, we used all samples with scATAC-seq to generate cluster visualizations of cell subpopulations with different chromatin-accessibility profiles (Fig. 1d) (see "**Methods**"). We identified clusters that were unique or shared across the samples (Fig. 1e). Unique clusters were specific to RES-A, RES-B, or both (named ENZ-induced clusters), or specific to the untreated and/or short-term ENZ-treated parental line (named initial clusters). Shared clusters were present at similar proportions across the samples and were named persistent clusters (Fig. 1e). We compared each cluster to all other clusters to determine its unique chromatin profile based on differentially accessible chromatin regions (DARs; Supplementary Data 1).

The most prevalent chromatin-based scATAC-seq clusters in terms of cell number (0, 1, and 2) were persistent (Fig. 1e) and defined by fewer than 20 unique DARs, suggesting that 74% of the cells share an overall similar chromatin-accessibility profile during the development of ENZ resistance. We then assessed for changes in cluster chromatin DARs between the parental LNCaP, LNCaP–ENZ48, and in RES-A and RES-B (Supplementary Data 1). DARs were observed around *MYC* and *TP53* in several clusters during the short-term response to enzalutamide, including in cluster 6 that arises during ENZ resistance in RES-A.

Prior studies have shown that PC cell lines cultured for an extended time without androgens tend to display neuroendocrine-like phenotypes[20,21]. The largest fold changes in chromatin accessibility based on average signal from all cells showed over representation for neuronal system processes between the parental (LNCaP or LNCaP–ENZ48) and resistant cells (RES-A or RES-B) (Benjamini–Hochberg-adjusted $p = 0.0027$ in RES-A and $p = 0.0024$ in RES-B, hypergeometric test). Accordingly, we found elevated expression of NEPC-derived signatures[20,22] in RES-A and RES-B cells (particularly *EZH2*, *AURKA*, *STMN1*, *DNMT1*, and *CDC25B*), as well as increased expression of NEPC-downregulated genes in initial clusters (Supplementary Fig. 1h). Interestingly, in bulk RNA sequencing of the same cell lines, gene set variation analysis (see "**Methods**") of NEPC signatures showed higher expression of NEPC-upregulated genes in RES-A cells only (Supplementary Fig. 1i).

 

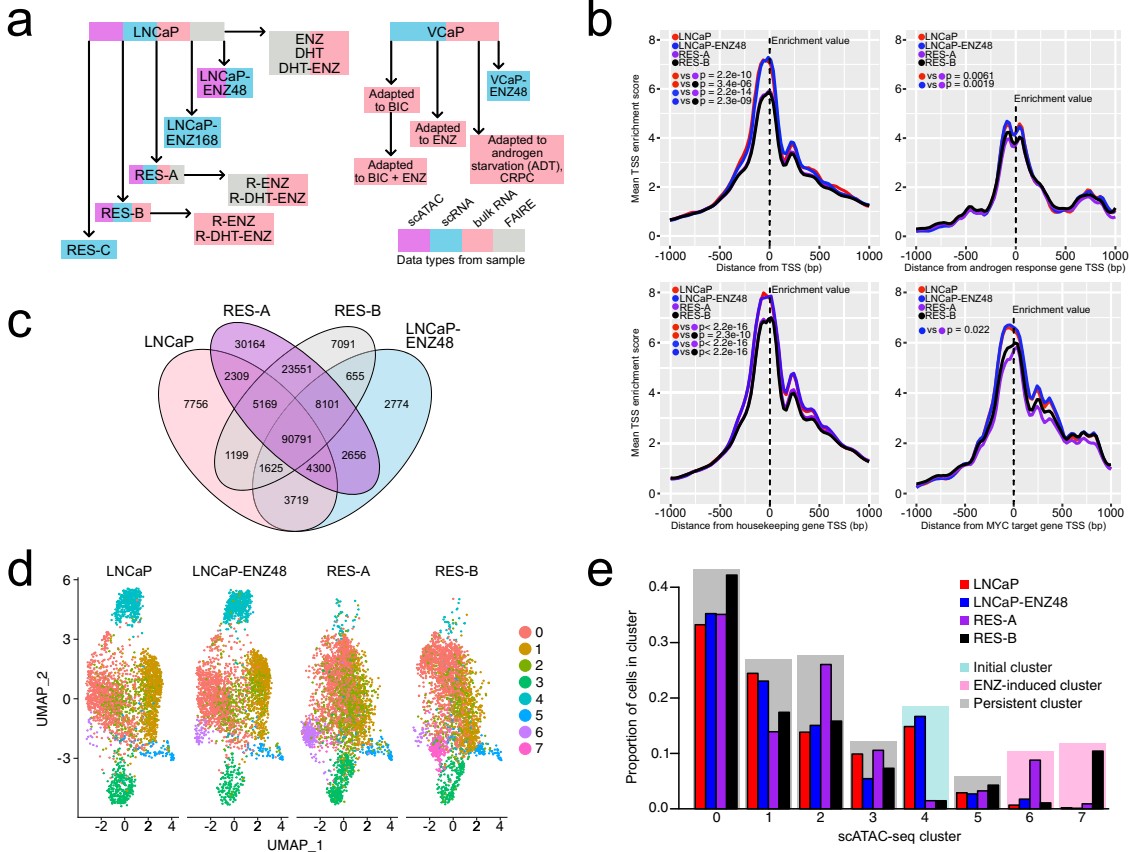

**Fig. 1 Chromatin reprogramming in enzalutamide resistance. a** Overview of the cell-line models, assays, and treatments included in the study. Boxes with sample names are colored according to the data types generated from the sample (single-cell (sc) assay for transposase-accessible chromatin (ATAC)-, scRNA-, bulk RNA- and/or formaldehyde-assisted isolation of regulatory element (FAIRE) sequencing). **b** Smoothed line plots of LNCaP parental, LNCaP–ENZ48, RES-A, and RES-B scATAC-seq enrichment scores in a 2-kb window around the transcription-start sites (TSS) of 5000 randomly selected genes, housekeeping genes, androgen response genes (MSigDB), and MYC target genes (MSigDB). Enrichment scores at each TSS (position 0 in the plot marked with a black dashed line) were used as the enrichment values and compared between pairs of samples. Sample comparisons are indicated using colored dots and the two-sided Wilcoxon rank-sum p-values are shown within the plots. **c** Venn diagram of shared and unique chromatin regions in LNCaP parental, LNCaP-ENZ48, RES-A, and RES-B based on a bulk analysis of scATAC-seq. **d** Uniform manifold approximation and projection (UMAP) scATAC-seq clustering visualization of LNCaP parental, LNCaP–ENZ48, RES-A, and RES-B. **e** Proportions of cells in the scATAC-seq clusters. Clusters are colored according to cluster type: initial (most prevalent in LNCaP parental and LNCaP–ENZ48), ENZ-induced (most prevalent in RES-A or RES-B), or persistent (present at similar proportions in all samples). ENZ = enzalutamide, DHT = dihydrotestosterone, BIC = bicalutamide, CRPC = castration-resistant prostate cancer. See also Supplementary Fig. 1.

Overall, these data show extensive chromatin reprogramming during the emergence of resistance to AR-targeting agents.

**Enzalutamide resistance reconfigures availability of TF binding DNA motifs in the chromatin.** Chromatin accessibility determines the transcriptional output of cells by exposing a footprint of TF DNA-binding motifs. We hypothesized that increased chromatin opening in resistant cells would change the footprint of exposed TF DNA motifs. To this end, we first utilized AR and MYC binding-site maps in LNCaP cells[23] and explored their relationship with open chromatin sites in the bulk FAIRE-seq data from RES-A cells. Using read-distribution analysis, we observed a significant increase in open chromatin at MYC-binding sites in ENZ-resistant cells ($p < 0.001$ in castrate conditions and with androgens, $t$-test) (Fig. 2a, Supplementary Fig. 2a), a reduction of open chromatin at AR-binding sites in castrate conditions ($p < 0.001$, $t$-test), and interestingly, an increase in open chromatin at AR-binding sites in androgen-exposed conditions ($p < 0.001$, $t$-test) (Fig. 2b, Supplementary Fig. 2b). These findings suggest that chromatin dysregulation in ENZ resistance is associated with reconfiguration of AR and MYC chromatin

binding, consistent with previously reported increased MYC and reduced AR transcriptional activity in these cells[16].

To resolve how chromatin reprogramming affects TF DNA motif exposure at the single-cell level, we performed a TF motif enrichment analysis on the marker DARs characterizing the scATAC-seq cell clusters in each sample (Fig. 2c). This analysis confirmed the enrichment of motifs for several PC-associated TFs such as AR and MYC, as well as GATA2, HOXB13, and others in persistent clusters 3 and 5, as well as initial cluster 4 in parental and LNCaP–ENZ48 (Fig. 2c). Clusters 3 and 5 remained enriched for a subset of the same TF motifs in RES-A and RES-B, with cluster 5 showing a consistent enrichment profile in all samples (Fig. 2c). AR, CREB1, E2F1, GATA2, and ZFX were common motifs. Cluster 3 was characterized by FOXA1 and JUND, while cluster 5 was characterized by CTCF, ETS-like, and MYC. Although they possessed distinct sets of DARs, the ENZ-induced clusters 6 and 7 did not display enrichment of TF motifs in RES-A or RES-B.

Between pairs of samples, DARs were predominantly closing and opening in cluster 0 compared with all other clusters (on average 43% of differentially closed DARs and 37% of

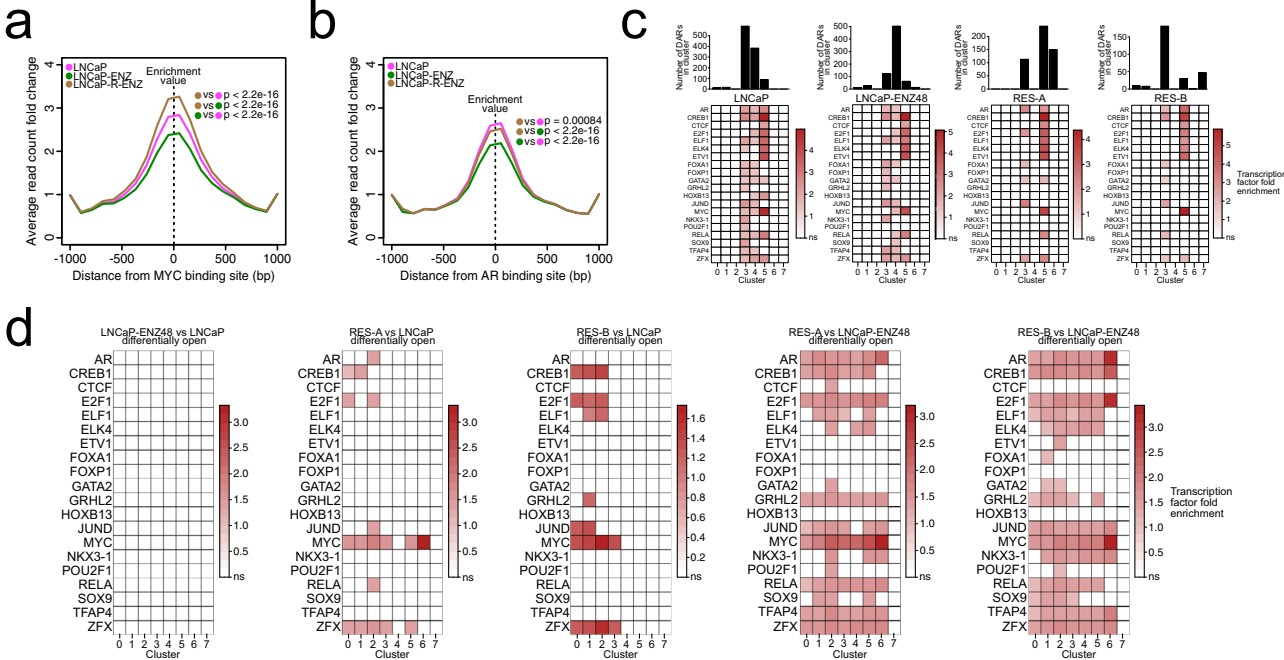

**Fig. 2 Contribution of enzalutamide treatment-mediated chromatin reprogramming to transcription factor DNA motif footprint. a, b** Normalized average formaldehyde-assisted isolation of regulatory element (FAIRE)-seq read distribution in androgen-deprived conditions within a 2-kb interval around **a** MYC-binding sites and **b** AR-binding sites in LNCaP cells. Sample comparisons of enrichment values at the middle of the distribution are indicated using colored dots within the plots and the two-sided *t*-test *p*-value is shown. **c** Prostate cancer-associated transcription factor (TF) motif enrichment in open differentially accessible regions (DARs) for each single-cell ATAC-seq sample. Enrichments with a Benjamini–Hochberg method-adjusted hypergeometric test *p*-value < 0.05 are shown in colors, while nonsignificant (ns) enrichments are shown in white. The barplots above the matrices indicate the number of open DARs found for each cluster in each sample. **d** TF motif enrichments in open DARs observed comparing the indicated conditions. Enrichments with a Benjamini–Hochberg method-adjusted hypergeometric test *p*-value < 0.05 are shown in colors, while nonsignificant (ns) enrichments are shown in white. See also Supplementary Fig. 2.

differentially open DARs) (Supplementary Fig. 2c, Supplementary Data 1). We performed selective TF motif enrichment analysis in DARs opened (Fig. 2d) and closed (Supplementary Fig. 2d) between pairs of samples (see "**Methods**"). While we observed no enrichments after short-term ENZ treatment (LNCaP–ENZ48 vs parental; Fig. 2d), comparing open DARs in RES-A or RES-B to the LNCaP parental retrieved distinct sets of TFs, with MYC and ESR1 being the most common across all clusters in RES-A and RES-B, respectively (Fig. 2d). Similarly, comparing open DARs in RES-A or RES-B vs LNCaP–ENZ48 showed enrichment of most of the PC-related TF motifs tested in most clusters (Fig. 2d), and to an even greater extent when considering closing DARs between sample conditions (Supplementary Fig. 2d).

These analyses demonstrate that ENZ resistance is associated with reconfiguration of TF DNA motif footprints.

**Transcriptional patterns of enzalutamide resistance are induced by divergent chromatin reprogramming.** To study transcriptional patterns in relation to reconfiguration of chromatin structure at the single-cell level, we performed scRNA-seq in the LNCaP parental, RES-A and -B models. Integrated clustering of four LNCaP samples (Fig. 3a) (see "**Methods**") showed 7 persistent, 3 ENZ-induced, and 3 initial cell clusters (Fig. 3b) defined by sets of marker differentially expressed genes (DEGs; Supplementary Data 2; between 17 and 283 DEGs in the 13 clusters). To confirm that these cell subpopulations are relevant in other independent models of ENZ resistance, we used the label-transfer approach[24] to query for matching cell populations in independent scRNA-seq datasets: a LNCaP parental sample, LNCaP ENZ treated for one week (LNCaP–ENZ168), and an

independent ENZ-resistant (RES-C) LNCaP-derived cell line (Fig. 1a). Transferring scRNA-seq cluster labels confirmed the presence of initial clusters (4, 6, and 10) in LNCaP parental (Supplementary Fig. 3a) and RES-C (Supplementary Fig. 3b). The presence of ENZ-induced clusters was confirmed in RES-C (17% of cells in cluster 3) and LNCaP-ENZ168 (79% in cluster 3), suggesting that one week of ENZ treatment is sufficient to give rise to this cluster prior to the development of resistance (Supplementary Fig. 3c). As a proportion of cells from most scATAC-seq clusters were additionally found to correspond to cluster-3 cells from the scRNA-seq (Fig. 3e), this suggests that the cells of this cluster may represent a common genomic configuration for ENZ resistance or its development. Most importantly, we could retrieve persistent subpopulations of cells in the alternative LNCaP-parental sample (4%), in LNCaP–ENZ168 (13%), and in RES-C (31%), suggesting that these persistent cells are consistently found during the emergence of ENZ resistance.

We additionally performed scRNA-seq on a VCaP parental cell line treated with DMSO or ENZ for 48 h to test for the generalizability of our results beyond a single-cell line (Fig. 1a). A similar analysis with VCaP cells confirmed the prevalence of persistent cells in the VCaP parental (93% of cells), as well as initial and ENZ-induced cells in VCaP–ENZ48 (38 and 55% of cells, respectively) (Fig. 3c).

We then sought to determine whether the observed scRNA-seq clusters (Fig. 3a) could be the result of enriched TF-binding activity in alternative open DARs. Using annotated databases, we queried the transcriptional targets of the enriched TFs in the open DARs when comparing RES-A or B to the parental LNCaP (Fig. 2d) in the matching scRNA-seq samples (see "**Methods**"). Chromatin remodeling affected TF activity and consequently

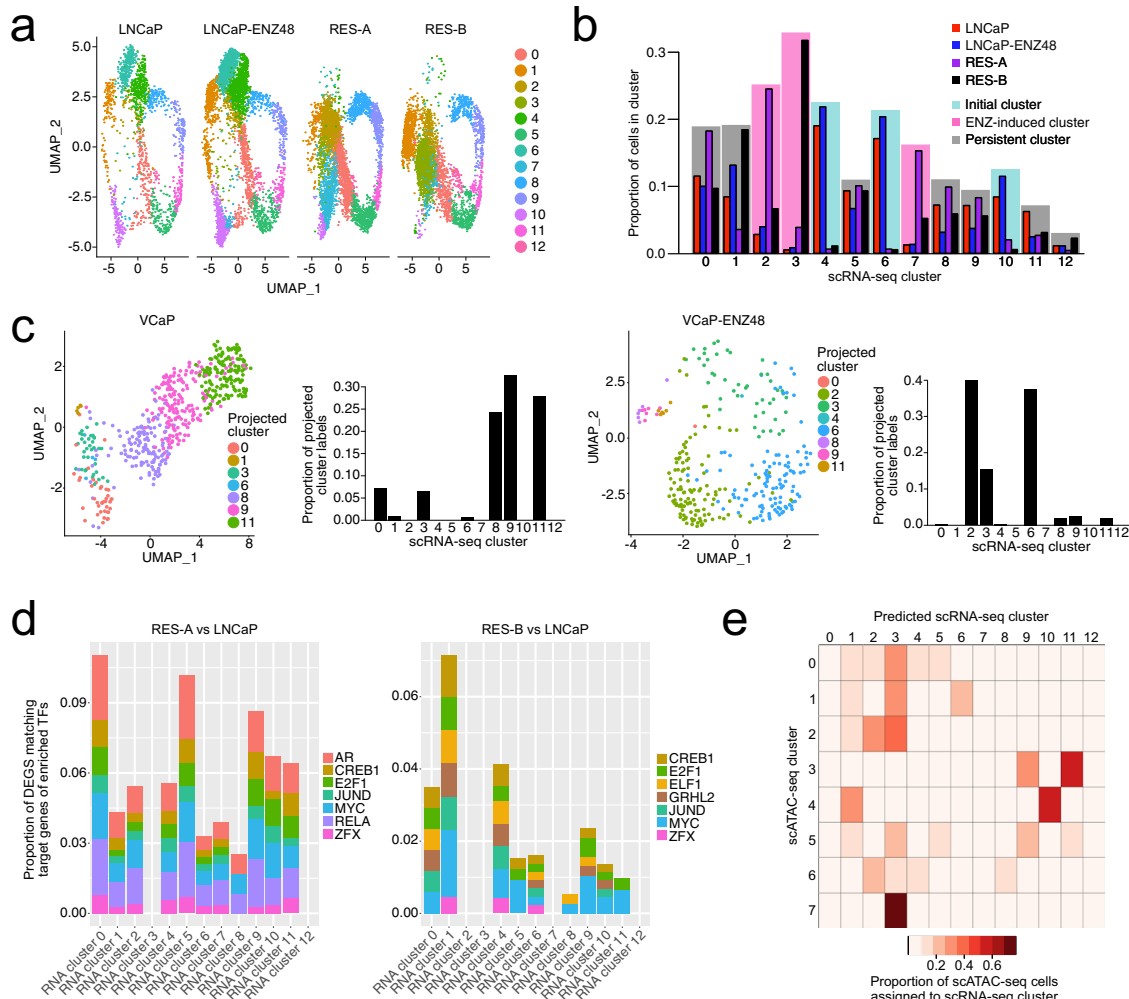

**Fig. 3 Chromatin states of enzalutamide resistance can result in multiple transcriptional programs. a** Uniform manifold approximation and projection (UMAP) clustering visualization of single-cell RNA sequencing (scRNA-seq) of LNCaP parental, LNCaP–ENZ48, RES-A, and RES-B. **b** Proportions of cells in clusters identified from scRNA-seq. Clusters are colored according to cluster type: initial (most prevalent in LNCaP parental and LNCaP–ENZ48), ENZ-induced (most prevalent in RES-A or RES-B), or persistent (present in similar proportions in all samples). **c** Cluster-label transfer from the integrated clustering of the LNCaP scRNA-seq data to VCaP parental (left) and VCaP treated with enzalutamide for 48 h (right), confirming the presence of these cell states in the alternate model. In the UMAP, each cell is colored according to the LNCaP scRNA-seq cluster that it is predicted to belong to. The barplot shows the proportion of the projected cluster labels for each scRNA-seq cluster. **d** Proportion of differentially expressed genes (DEGs) in each LNCaP scRNA-seq cluster for the indicated sample comparisons that is composed of enriched transcription-factor (TF) target genes. The contributions of enriched TFs identified in the scATAC-seq are shown as a stacked barplot. **e** Identification of matching cell clusters between the scRNA- and scATAC-seq data from LNCaP visualized as heatmap. The heatmap shows the proportions of cells from each scATAC-seq cluster across all sample conditions assigned to each scRNA-seq cluster as part of the label-transfer process. The proportions were calculated for each scATAC-seq cluster, with the total as the number of cells from the scATAC-seq that could be confidently assigned to a scRNA-seq cluster (confidence score > 0.3). ENZ = enzalutamide. See also Supplementary Fig. 3.

DEGs in the scRNA-seq for up to a maximum of 11% in cluster 0 in RES-A and 7.1% in cluster 1 in RES-B (Fig. 3d). While target DEGs for TFs such as MYC, JUND, and E2F were found in most clusters in both RES-A and -B, other target DEGs for TFs such as AR, RELA (a NF-kB subunit), and GRHL2 appeared more specific to RES-A or -B, consistent with the proposed stoichiometric models of TF chromatin binding[25]. This analysis confirmed that alternative open DARs in ENZ resistance can activate divergent transcriptional programs.

Next, we connected the scRNA-seq clusters to their matching scATAC-seq clusters. We again leveraged the label-transfer approach to identify matching scRNA- and scATAC-seq cell states in the same sample conditions (see "**Methods**"). In this

process, we assigned cell cluster labels within the scRNA-seq sample to the scATAC-seq cells, or vice versa. We found that a chromatin state can correspond to multiple transcriptional states (96% of cells mapped from scATAC to scRNA vs 28% of cells mapped from scRNA to scATAC in parental LNCaP, $p < 2.2e{-}16$, chi-square test) (Supplementary Fig. 3d). By querying the integrated scRNA-seq clusters in the scATAC-seq data (Fig. 3a, b), we could find matching cell states in the scRNA for all scATAC clusters, with cells in an scATAC cluster generally corresponding to multiple scRNA clusters (Fig. 3e, Supplementary Fig. 3e). Notably, 58% of scATAC-seq cluster-4 cells were projected to belong to scRNA-seq cluster 10, 83% of scATAC-seq cluster-3 cells were predicted to belong to scRNA-seq cluster 9 or

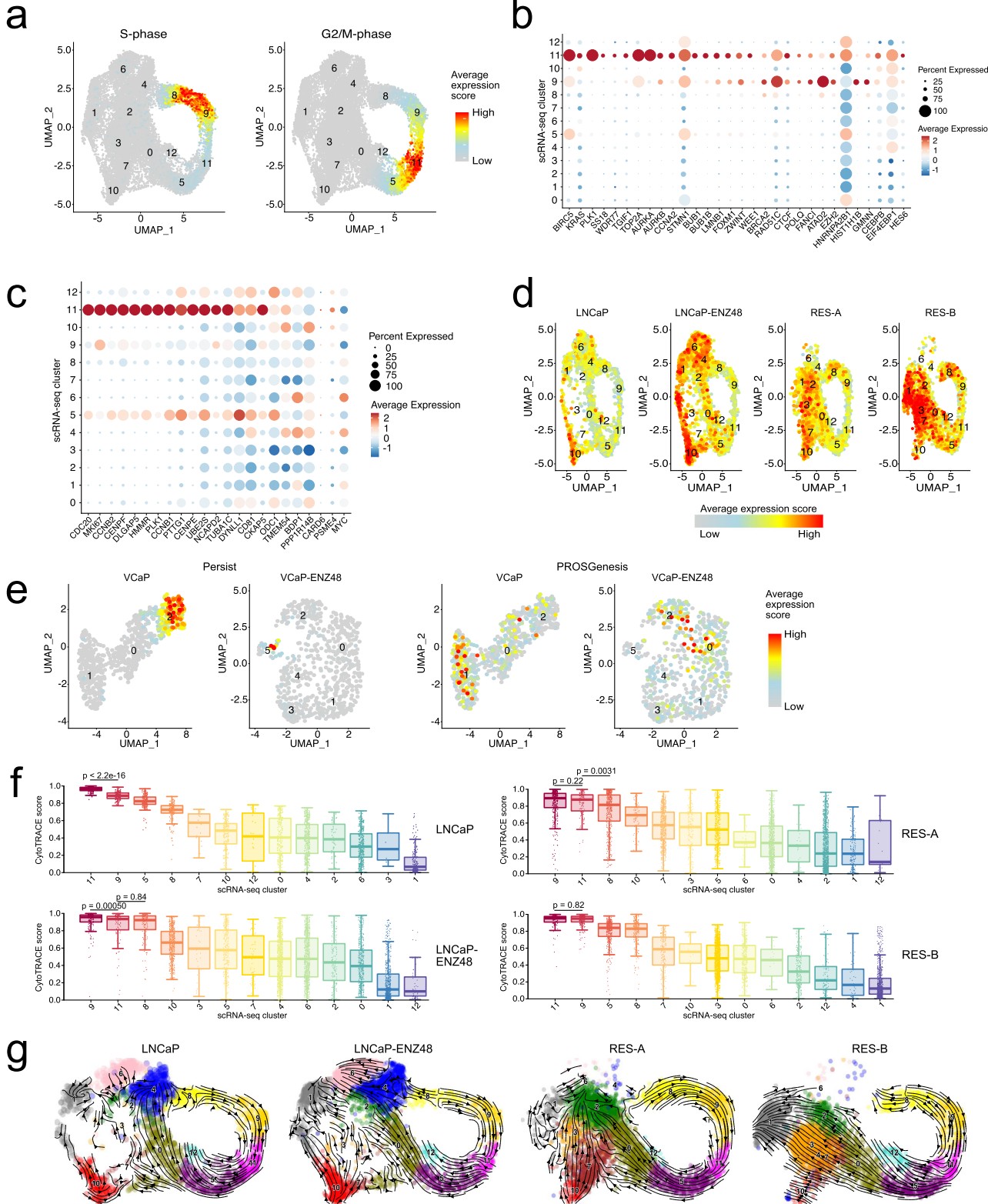

11, and 77% of scATAC-seq cluster-7 cells were projected to belong to scRNA-seq cluster 3 across the sample conditions (Fig. 3e).

Taken together, these data show that transcriptional configuration of ENZ-resistant cells, especially cells persisting during treatment, emerges from processes driven partially by chromatin structure and TF-mediated transcriptional reprogramming. These processes affect a number of important regulators of cell fate,

consistent with lineage commitment recently observed in tissue development[26].

**Prostate cancer cell subpopulations with features of stemness precede enzalutamide resistance**. Cell cycle phase can be a strong determinant of the integrative clustering of scRNA-seq data. Accordingly, we found that persistent clusters 8, 9, and

**Fig. 4 Transcriptional states in enzalutamide resistance. a** Average expression of cell cycle-related genes (S- and G2/M phases) in cells from the single-cell (sc) RNA-seq data. **b, c** Dot plot of average gene expression of the **b** indicated genes and of the **c** genes within the Persist signature in each scRNA-seq cluster. The size of the dot reflects the percentage of cells in the cluster that express each gene. **d** Uniform manifold approximation and projection (UMAP) visualization showing the average expression score of each cell for the genes in the PROSGenesis gene signature derived from Karthaus et al.[30]. **e** Cells in VCaP and VCaP–ENZ48 (enzalutamide treated for 48 h) scored for their expression of Persist and PROSGenesis gene signatures. **f** Boxplots of predicted cluster-differentiation states in the four LNCaP scRNA-seq samples based on cytoTRACE. Each cell is visualized as a point within its scRNA-seq cluster. Clusters are ordered from left to right in order of decreasing predicted differentiation potential. The scRNA-seq clusters are labeled with numbers. The boxplots show the 25th percentile, median, and 75th percentile, with the whiskers indicating the minimum and maximum values within the 1.5x interquartile range. The two-sided Wilcoxon rank sum test was used to assess for differences in average cluster cytoTRACE scores and p-values are shown within the figure (n = 111, 106, 122, and 126 for cluster-11 cells in LNCaP, LNCaP-ENZ48, RES-A, and RES-B; n = 127, 160, 379, and 226 for cluster-9 cells in LNCaP, LNCaP-ENZ48, RES-A, and RES-B; and n = 135, 451 for cluster-8 cells in LNCaP–ENZ48 and RES-A). **g** RNA velocities based on scRNA-seq depicted as streamlines. Clusters are shown in different colors and are numbered. ENZ = enzalutamide. See also Supplementary Fig. 4.

11 scored highly for S- and G2/M-phase-related genes using cell cycle scoring in Seurat (see "**Methods**") (Fig. 4a), suggesting that cells in these clusters are more actively cycling and proliferating. However, we found that cells in cluster 11 were characterized not only by cell cycle genes, but also by the expression of genes involved in chromatin remodeling and organization (*CTCF*, *LAMINB*, *ATAD2*, and *SS18*), regulation of cell proliferation and stemness (*HES6* and its target *PLK1*[27], *KRAS*, *FOXM1*, and its targets *BIRC5*, *AURKA*, *AURKB*, and *CCNA2*[28]), and DNA repair (*BRCA2*, *FANCI*, *RAD51C*, and *POLQ*) (Fig. 4b). Clusters 5 and 11 showed high expression of a gene set that characterized a cell subpopulation with stem-like, androgen-insensitive, and cell cycle-driven features from Horning et al.[29] (Fig. 4c). We named this gene signature Persist to highlight its association with cluster 11, a cell state that persists despite exposure and resistance to ENZ. Karthaus et al. recently identified activated luminal prostate cells able to regenerate the epithelium following castration[30]. We extracted the gene expression profile associated with these prostate luminal cells (see "**Methods**") and used it to score each scRNA-seq cluster. We found the initial cluster 10 to score highly for this gene signature prior to ENZ treatment in the parental LNCaP and named it PROSGenesis (Fig. 4d, Supplementary Fig. 4a). We visualized the expression of PROSGenesis and Persist genes in scRNA-seq samples from VCaP to confirm the presence of these subpopulations of cells in other models (Fig. 4e).

We then set out to reconstruct the trajectories of how these clusters of interest arise during the development of ENZ resistance. Using CytoTRACE[31] estimation of differentiation states based on the number of expressed genes, cells in cluster 11 showed high developmental potential in most sample conditions (Fig. 4f), suggesting that other cell subpopulations could derive from cells in this cluster. RNA velocity analysis estimated cluster 10 as a precursor of the enzalutamide-induced clusters (Fig. 4g), concordant with a state derived from activated regenerative luminal prostate cells as previously suggested[30]. Cluster-specific differential velocity analysis in RES-A and RES-B revealed downregulation of many PC-related genes such as *ATAD2*, as well as upregulation of genes such as *UBE2T*, *PIAS2*, *PFKFB4*, and *EGFR* (Supplementary Fig. 4b, c). *ATAD2* and *UBE2T* were otherwise upregulated in persistent clusters 8, 9, and 11 (Supplementary Fig. 4c), suggesting that additional transcriptional reprogramming occurs in the ENZ-induced clusters.

These analyses point at two distinct subpopulations of PC cells that precede ENZ resistance: one persistent cell cluster (cluster 11) matching Persist and one initial cluster (cluster 10) matching PROSGenesis, a signature derived from tissue regeneration[30]. Collectively, our data suggest that a small number of PC cells with regenerative potential exist within the bulk tumor.

**Model-based characterization of gene signatures in prostate cancer bulk RNA sequencing.** The use of molecular gene classifiers or signature scores is an attractive strategy to select cancer patients for treatment[13,32]. According to gene set variation analysis, most of the persistent clusters and cluster 10 showed enrichment of E2F target, G2M checkpoint, and MYC target genes (Supplementary Fig. 4d). These data are largely concordant with the bulk RNA-seq data on the same cells in our previous study[16], demonstrating that signals from subpopulations of cells can be retrieved in bulk RNA-seq data. Differential expression within clusters (Supplementary Data 2) and gene set enrichment analysis further revealed that oxidative phosphorylation was immediately upregulated in LNCaP–ENZ48, and that this process is maintained selectively in RES-A but not in RES-B cells (Supplementary Fig. 4e–g). Moreover, genes regulated by activated mTORC1 signaling were consistently upregulated in most of the clusters during the development of ENZ resistance (Supplementary Fig. 4e–g), in agreement with previous reports showing its activation during ENZ treatment in patients[26].

We therefore used a collection of signatures derived from the scRNA-seq analysis to describe features of the same cells in bulk RNA-seq datasets. In addition to Persist and PROSGenesis, we included (1) NEPC markers (Supplementary Fig. 1g), (2) a BRCAness gene signature[33] as RES-A and RES-B maintain sensitivity to PARP inhibition[16] and the persistent cluster 11 is characterized by markers of DNA repair (Figs. 4b), (3) gene sets as proxies of AR signaling activation[3], including activation of AR-splice variants (AR-Vs), (4) the DEGs defining our scRNA-seq clusters, and (5) gene sets for mTORC1 signaling and MYC targets (Supplementary Fig. 4d–g) (Supplementary Data 3).

In the bulk, the ENZ-induced DEGs selectively appeared in the RES-B cells (Fig. 5a). Similarly, the persistent clusters were associated with the Persist signature only in RES-A and RES-B when induced with DHT (Fig. 5a). On the other hand, the PROSGenesis signature was elevated only in RES-B (Fig. 5a).

To confirm the properties of different signatures, we used VCaP cells to develop an independent model of resistance to AR signaling-targeted treatments, including ADT, bicalutamide, ENZ, and bicalutamide/ENZ multiresistant sublines, and performed bulk RNA-seq (Fig. 1a). These VCaP-based sublines did not show NE features (Fig. 5b). Only ENZ-resistant VCaP cells scored highly for the ENZ-induced DEGs, confirming the specificity of this signature to ENZ treatment and resistance. Parental and ENZ-resistant VCaP cells scored highly for the PROSGenesis signature, while the scores of the persistent, Persist, mTORC1 signaling, and MYC target signatures scored highly selectively in resistant VCaP sublines (Fig. 5b). This suggests a convergent mechanism of resistance to these agents in independent models.

Next, we scored xenografts of AR⁺/NE⁻, AR⁻/NE⁺, or AR⁻/NE⁻ CRPC and NEPC tumors resistant to ENZ[34,35] with the same signature set (Supplementary Fig. 5a). AR⁺/NE⁻ xenograft samples clustered into two separate clusters. AR⁻ tumors clustered together

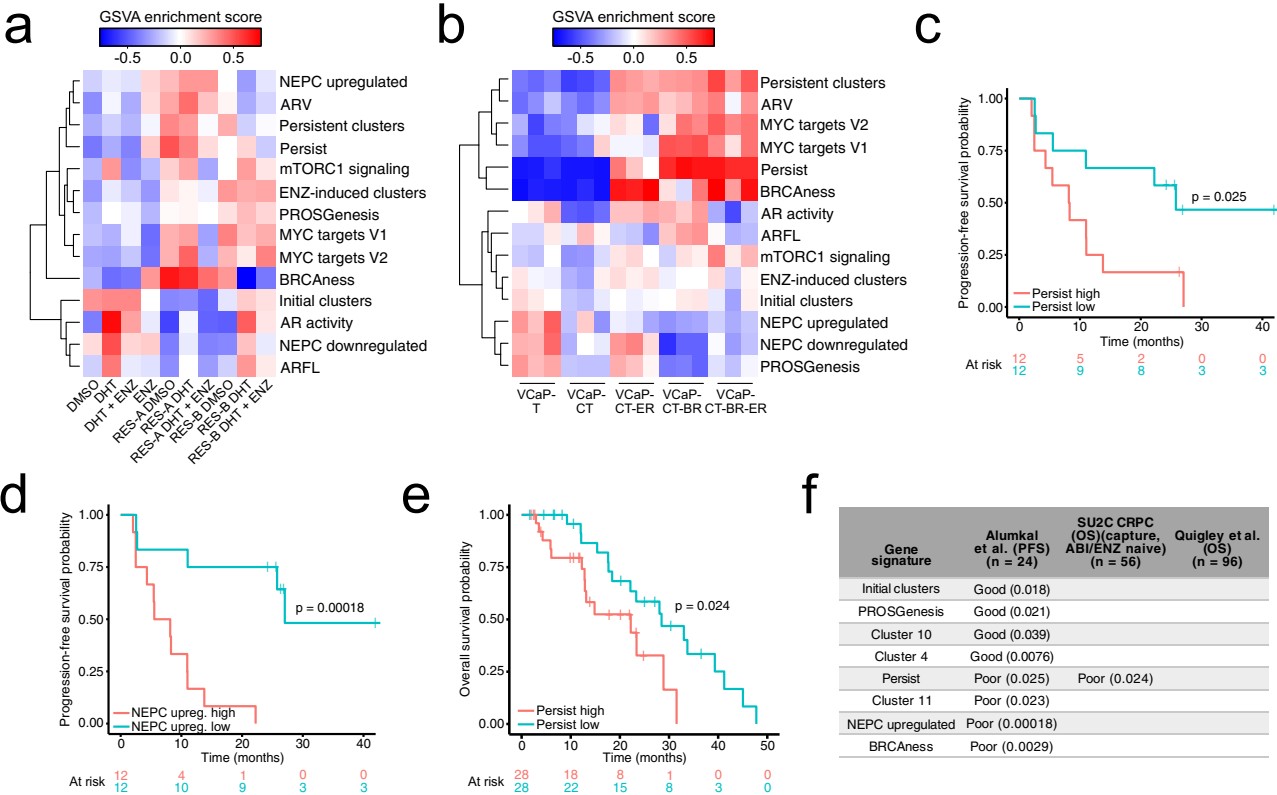

**Fig. 5 Gene signatures derived from single-cell RNA sequencing capture important features of prostate cancer models and stratify patients with advanced disease. a** Heatmap of single-cell gene signature gene set variation analysis (GSVA) enrichment scores in bulk RNA-sequencing of LNCaP treated with dihydrotestosterone (DHT) or enzalutamide (ENZ), and either sensitive or resistant to ENZ. **b** Heatmap of gene signature GSVA enrichment scores in bulk RNA sequencing from VCaP subline derivatives VCaP-T (long-term cultured with 10 μM testosterone), VCaP-CT (VCaP-T long term cultured with 0.1 nM testosterone), VCaP-CT-ET (VCaP-CT cultured long-term with 10 μM ENZ), VCaP-CT-BR (VCaP-CT cultured long term with bicalutamide), and VCaP-CT-BR-ER (VCaP-CT-BR long-term treated with ENZ upon reaching bicalutamide insensitivity). **c** Kaplan–Meier progression-free survival curves for Alumkal et al.[2] patients stratified into two groups based on median GSVA score for the Persist gene signature. The two-sided log-rank p-value is shown above the curves. **d** Kaplan–Meier progression-free survival curves for Alumkal et al. patients stratified into two groups based on median GSVA score for the NEPC-upregulated gene signature. The two-sided log-rank p-value is shown above the curves. **e** Kaplan–Meier overall survival curve for abiraterone (ABI)- and ENZ-naive patients from the Stand Up 2 Cancer (SU2C) CRPC cohort[5] stratified into two groups based on median GSVA score for the Persist gene signature. The two-sided log-rank p-value is shown above the curve. **f** Summary table of gene signature GSVA score associations with progression-free survival (PFS) or overall survival (OS) in the clinical datasets. Only gene signatures significantly associated with PFS or OS in one or more datasets are shown. Good indicates that a higher score for the signature (a score higher than the median) is associated with better survival outcome, while poor indicates that a higher signature score is associated with worse survival outcome. Two-sided log-rank p-values are shown in parentheses. For each dataset, the header indicates the number of samples included, along with other qualifying information of the dataset. We used ABI/ENZ-naive patients from the capture-based SU2C CRPC RNA-seq dataset. See also Supplementary Fig. 5.

with a series of AR$^+$/NE$^-$ tumors due to low mTORC and MYC signaling, while one cluster of AR$^+$/NE$^-$ scored highly for all gene sets apart from genes upregulated in NEPC. Interestingly, the PROSGenesis signature, along with initial clusters and ENZ-induced clusters, scored particularly high in AR$^+$ tumors, while the Persist signature, along with the persistent clusters, scored high in both AR$^+$/NE$^-$ and AR$^-$/NE$^+$ tumors (Supplementary Fig. 5a). This suggests that the two signatures capture different tumor biologies. In a transcriptome dataset based on an independent xenograft model[36], we found ENZ resistance to be uniquely associated with higher AR activity, higher expression of MYC target genes, high score for PROSGenesis, and high expression of the ENZ-induced cluster gene set (Supplementary Fig. 5b). These data indicate that Persist status is independent of AR status and that persistent cells might mediate the development of both AR-positive CRPCs and AR-negative NEPCs.

Collectively, the persistent cluster, initial cluster, PROSGenesis, and Persist gene signatures show potential for identifying aggressive, regenerative features of PC from bulk RNA-seq.

**Transcriptional signal enrichment analysis identifies treatment-persistent cells and prognostic gene signatures in prostate cancer patients.** We hypothesized that we could use enrichment of gene signature expression to stratify advanced and primary PC patients.

To this end, we interrogated clinical data of CRPC patients treated with ENZ reported in Alumkal et al.[2] The patients aggregated into two clusters based on our complete signature set (Supplementary Fig. 5c), but patients in neither cluster had significantly shorter progression-free survival (PFS; p = 0.049, log-rank test). Utilizing a stepwise variable-selection process we identified five significant signatures (NEPC upregulated, PROS-Genesis, MYC targets, AR activity, and ARV) that are able to identify patients with significantly shorter PFS (Supplementary Fig. 5d). Moreover, PFS analysis of individual gene signatures revealed association with shorter time to progression for patients scoring high for the Persist signature (p = 0.025, log-rank test) or for genes upregulated in NEPC (p = 0.00018, log-rank test)

(Fig. 5c, d), while patients with longer PFS scored highly for PROSGenesis ($p = 0.021$, log-rank test) and for the initial cluster signature ($p = 0.018$, log-rank test) (Supplementary Fig. 5e).

None of the cluster marker gene sets showed a significant difference between Stand Up To Cancer (SU2C) CRPC abiraterone/ENZ-naive and abiraterone/ENZ-exposed patients[5] according to their latest treatment regime. This suggests that it may be difficult to retrieve differences between these bulk-sequenced tumors from heavily pretreated patients using single-cell-derived signatures. However, Persist was still significantly associated with poor overall survival in these patients (Fig. 5e), supporting the potential significant activity of the persistent cells in this group of patients. Similarly, we could not stratify patients that developed resistance to ENZ in the SU2C West Coast DT Quigley et al. dataset[37] (Supplementary Fig. 5f), although in this case, ENZ-sensitive patients had higher expression of PROSGenesis ($p = 0.024$, Wilcoxon rank-sum test) (Supplementary Fig. 5g).

These data show that the Persist signature associated with persistent cells (cluster 11) from our single-cell analysis of ENZ resistance is a consistent classifier with the potential of stratifying patients for response to second-line AR-targeted treatments (Fig. 5f).

We then hypothesized that we could systematically use the persistent cluster 11, Persist, initial cluster 10, and PROSGenesis signatures as a proxy for the presence of PC cells with different transcriptional features in clinical settings and to capture signals from pre-existing subclones with metastatic potential in primary untreated tumors. To this end, we took advantage of a recently published scRNA-seq dataset on clinically relevant PC specimens[38] (Fig. 6a). We used GSVA scoring to highlight our 13 scRNA-seq clusters in 36424 cells from 13 primary untreated PC specimens (Supplementary Fig. 6a). The analysis showed that our LNCaP model-derived cell clusters scored higher in luminal and basal/intermediate cells compared with fibroblasts ($p = 0.047$, $t$-test) (Supplementary Fig. 6a). Additionally, luminal cells had higher expression of genes associated with our initial scRNA-seq clusters compared with the basal/intermediate cells ($p = 0.020$, $t$-test) and compared with fibroblasts ($p = 0.00015$, $t$-test).

We then scored the cells for the expression of genes from the Persist and PROSGenesis signatures, along with their associated clusters (11 and 10, respectively) and control signatures linked to AR activity (ARV, AR-FL, and AR activation), BRCAness, and NEPC (Fig. 6b). We defined a high score for a gene signature to be above the 90th percentile. About 48% percent of the cells that scored highly for the Persist signature were luminal cells (Fig. 6c). Cells scoring highly for the PROSGenesis signature were mostly basal/intermediate (78% of high-scoring cells) (Fig. 6d). Each tumor harbored on average 8% of cells scoring high for the Persist signature (ranging from 2% in tumor 173 to 23% in tumor 156) and 8% of cells scoring high for the PROSGenesis signature (ranging from 0.9% in tumor 153 to 33% in tumor 172) (Fig. 6e).

To reconcile the presence of these cells and their relative histopathological position, we assessed gene expression within two sections of primary untreated PC (prostate A and B) with spatial transcriptomics (see "**Methods**"). We reconstructed the gene expression signal from stromal and epithelial components in an average of 1682 spots per sample using clustering analysis and annotated the tissue architecture in 5 clusters of stromal tissue (ST), benign epithelium (BE), and adenocarcinoma (PC–AC) (Fig. 6f, Supplementary Fig. 6b). PROSGenesis and Persist signatures, as well as the companion model-derived cluster 10 signature, showed high expression scores within the sections compared to scores from housekeeping gene signatures (Fig. 6g, Supplementary Fig. 6c). We compared the score distributions of our signatures to the housekeeping gene set score distributions

and determined the 90th percentile as a score cutoff for high expression by allowing for 5% false positives (see "**Methods**"). Spots with high signal were found interspersed in all five clusters in both sections (Fig. 6h, Supplementary Fig. 6d). In prostate A, however, spots scoring highly for the Persist signature were more prevalent in the PC–AC cluster compared with ST ($p = 0.0049$, chi-square test). Spots scoring highly for PROSGenesis were further enriched in the BE and PC–AC clusters compared with ST ($p < 0.001$ in both cases, chi-square test), while spots scoring highly for cluster 10 were enriched in the BE clusters compared to all other tissue regions ($p < 0.001$ for each comparison, chi-square test) (Fig. 6h). Concordant observations were made for PROSGenesis and cluster 10 signatures in prostate B (Supplementary Fig. 6d). To validate these findings, we undertook a similar approach to reanalyze spatial transcriptomics data from prostate sections 3.3, 1.2, and 2.4 from Berglund et al.[39], which were annotated to contain a significant proportion of cancer. Similar to our initial observations, these sections showed enrichment of spots scoring highly for the PROSGenesis in the PC–AC clusters compared with ST or prostatic intraepithelial neoplasia clusters (Supplementary Fig. 6e–g). Spots scoring highly for cluster 10 were more prevalent in both benign and cancerous clusters. Taken together, these data suggest the presence of treatment-persistent cells interspersed within the primary untreated prostate tissue of PC patients with high metastatic potential.

Finally, we verified whether we could predict recurrence in primary PC patients using the signature genes derived from these cells. We interrogated legacy primary tumor TCGA PRAD (https://www.cancer.gov/tcga) (Fig. 7a) and early-onset PC (EOPC) ICGC[12] RNA-seq data (Supplementary Fig. 7a) for our gene signatures of interest. Using all signatures for clustering, the TCGA PRAD cohort separated 54% of Gleason score (GS)-7 and 15% of GS-8+ patients that would not benefit from additional treatment, as they had relatively good prognosis (Fig. 7b). A similar trend was also observed in the ICGC cohort (Supplementary Fig. 7b). ENZ-induced cluster (Fig. 7c), PROSGenesis (Fig. 7d), Persist (Fig. 7e), and persistent cluster (Fig. 7f) gene signatures were the most significant contributors to cluster separation in the TCGA cohort, while NEPC-downregulated genes were the major determinant in the ICGC cohort (Supplementary Fig. 7c). In line with previous reports[2], signatures reflecting AR activity (AR activity and full length AR) in these tumors were consistently associated with longer time to progression in the TCGA cohort (Fig. 7g, h), suggesting a better response to inhibition of AR signaling in AR-driven tumors. In the EOPC cohort, which is enriched in GS-7 tumors compared with the TCGA PRAD cohort, the persistent cluster and PROSGenesis signatures significantly stratified GS-7 patients ($p = 0.034$ and $p = 0.021$, log-rank test) (Supplementary Fig. 7d, e), suggesting the ability of these signatures to further refine GS-based risk stratification in patients and avoid overtreatment. High PROSGenesis score was associated with good prognosis together with the gene set from the initial cluster 10 (Fig. 7i). Individually, 8 out of 13 cluster-derived signatures showed association with PFS in the TCGA cohort (Fig. 7i), pointing at the utility of these signatures in PC patient risk stratification.

## Discussion

In this study, we provide a molecular perspective on the emergence of resistance to AR-targeted treatment at a single-cell level. Karthaus and colleagues recently found that luminal prostate cells that persist after ADT in a mouse model can contribute to tissue regeneration of the normal prostate epithelium by assuming stem-like transcriptional properties[30]. Using PC specimen tumor

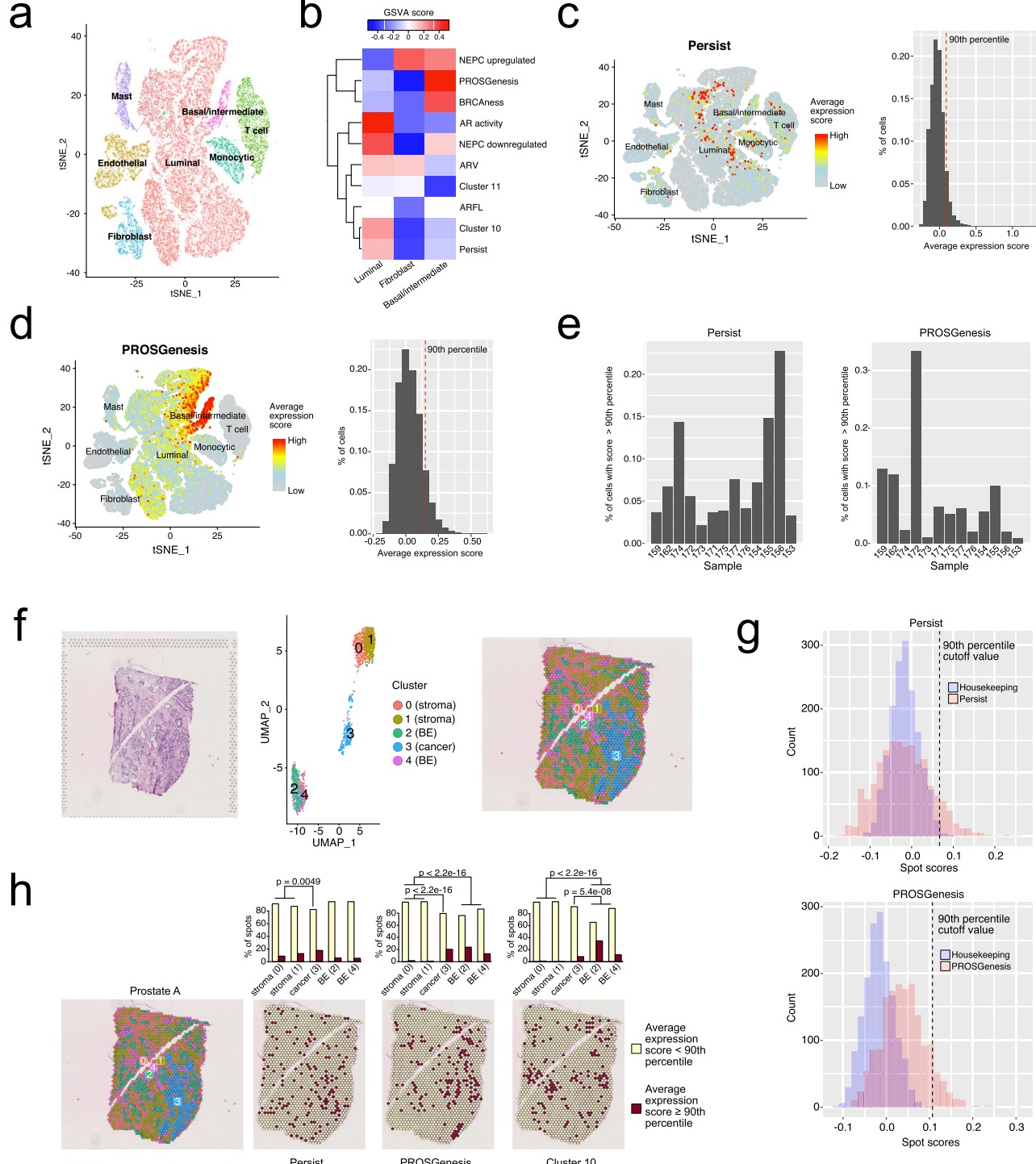

DNA, we recently showed the presence of subclones within primary tumors that preserve the ability to expand and metastasize years after treatment and that are found interlayered within different lesions of multifocal tumors[9]. Similarly, a recent work studying lung cancer metastases found that metastatic capacity arises from pre-existing and heritable differences in gene expression[40]. Here we find that during exposure to AR-targeting agents, a small proportion of persistent cells remain transcriptionally unperturbed by the treatment.

We visualize these cells in primary untreated PC specimens and find them to be enriched in cancerous regions of

histopathologically relevant tumors using spatial transcriptomics, as well as interspersed in apparent benign tissue. As differentiating between benign basal epithelia and tumor epithelia remains one of the major questions of the PC field, understanding the presence and function of these persistent and potentially regenerative cells in histopathologically noncancerous regions warrants further studies. Our data show evidence of a hierarchical model of emergence of enzalutamide resistance[41] in which treatment-persistent cells are able to regenerate the bulk of resistant cells. We describe the properties of the persistent cells using RNA velocity and show different intermediate states in

**Fig. 6 Transcriptional signal enrichment analysis identifies treatment-persistent cells in prostate cancer. a** $t$-Distributed stochastic neighbor embedding (tSNE) visualization of cell types from 13 treatment-naive prostate tumors from Chen et al.[38] **b** Gene set variation analysis (GSVA) enrichment scores for gene signatures in luminal, basal/intermediate, and fibroblast cells from Chen et al. GSVA enrichment scores were generated from the average expression profile of each cell type. **c–d** tSNE plot of prostate tumor cells from Chen et al. colored according to their average expression of the genes in **c** the Persist signature and **d** the PROSGenesis signature. The adjacent histograms show the distribution of average expression scores in the cells, with a red dashed line marking the 90th percentile of scores. **e** Percentage of cells scoring at or above the 90th percentile for the Persist and PROSGenesis signatures belonging to each prostate tumor from Chen et al. **f–h** Spatial transcriptomics (ST) from a prostate cancer tissue section, Prostate A. **f** The left panel shows the hematoxylin and eosin (H&E) staining of the tissue section. In the middle, the uniform manifold approximation and projection (UMAP) visualization shows the clusters of spots on the ST slide. Each cluster is labeled according to its histological tissue type, with BE referring to benign epithelium. The right panel shows the UMAP clusters of spots overlaid on the H&E slide. **g** Sensitivity analysis of Persist and PROSGenesis signatures scores in ST against the score distributions of control housekeeping gene signatures (see "**Methods**"). **h** The leftmost panel shows the ST UMAP clusters of spots overlaid on the H&E slide. Each spot was scored according to its expression of genes in the Persist, PROSGenesis, and cluster-10 signatures. For each signature, spots scoring at or above the 90th percentile (high) are colored in red, while spots scoring below the 90th percentile (low) are colored in yellow. The barplots indicate the percentage of spots in each cluster scoring high or low for each signature. The bars are labeled with the cluster histology and the cluster number in parentheses, with BE referring to benign epithelium. Differences in proportions of high-scoring spots between clusters were tested with the chi-square test. See also Supplementary Fig. 6.

alternative trajectories of treatment resistance. This process is partially driven by chromatin remodeling, which is consistent with chromatin-accessibility lineage priming[26,42].

In PC, gain of function of bromodomain-containing proteins such as BRD4[43,44] and ATAD2[44,45], as well as loss of function of the chromatin remodeler CHD[14], have been shown to contribute to PC progression and lineage plasticity in therapy resistance. This process is likely accompanied by chromatin reprogramming[20,44,46]. While many groups have focused on the effect of AR-targeted treatment on chromatin-associated factors such as CREB5[47] or TFs such as GR[48] and AR[49], in this study, we found that exposure to AR-targeting agents increases the overall relaxation of the chromatin. While we used separate assays for scATAC-seq and scRNA-seq in different cells, we were able to integrate these data using the label-transfer method across datasets. These analyses revealed that subpopulations of cells with different chromatin states may lead to multiple transcriptional configurations, including those of persistent cells. Availability of chromatin accessibility and gene expression data from the same cell would further reduce technical variation and enable more in-depth characterization of these configurations. Using cell-line models mimicking alternative trajectories of treatment resistance, we infer that differential DNA motif exposure determined by chromatin structure may partially contribute to TF activity-mediated transcriptional reprogramming in the different cell subpopulations induced by enzalutamide exposure. According to this analysis, specific subpopulations of PC cells are more subject to TF activity reprogramming than others. This is consistent with recent studies showing simultaneous detection of multiple transcription factors on single DNA molecules and TF cooccupancy frequently occurring at sites of competition with nucleosomes[18].

We show that treatment-persistent cells have high cell cycle turnover, compatible with high regenerative potential[50,51], and identify the transcriptional features characterizing this cell population. As these features have been associated with more aggressive tumors, we developed transcriptional signatures based on two cell states: one state renamed PROSGenesis derived from ADT-treated prostate cells by Karthaus et al.[30] and another state that we called Persist that was associated with persistent cells during the emergence of ENZ resistance. PROSGenesis, Persist, and associated signatures can capture different tumor types and stratify ARSI-exposed CRPC patients' outcome. Moreover, we show that in primary PC patients undergoing ADT treatment, high signature scores in treatment-naive specimens are associated with shorter time to progression (biochemical recurrence). Interestingly, in primary treatment-naive patients, high score for PROSGenesis is associated with longer response to ADT, possibly due to the stronger contribution of AR activity in these tumors.

Overall, we have identified and characterized gene signatures that can be used to profile subpopulations of treatment-persistent cells with regenerative properties that foster alternative trajectories of AR-targeted treatment-resistant PCs.

## Methods

**Cell lines and culture.** LNCaP and VCaP cell lines were obtained from American Type Culture Collection (ATCC; LGC Standards), authenticated periodically (HPA cultures or Eurofins), and tested for mycoplasma contamination monthly. RES-A and RES-B cells were generated by prolonged exposure to the second-generation anti-androgens enzalutamide and RD-162. RES-A was generated by passaging LNCaP cells in increasing concentrations of enzalutamide for 9.5 months, whereas RES-B was generated by continuous treatment with 10 μM RD-162 for 13 months[16]. LNCaP parental (ATCC), RES-A, and RES-B cells were cultured in RPMI 1640 (Sigma R0883) supplemented with 10% FBS (Sigma F7524), 2 mM Alanyl-glutamine (Sigma G8541), 1 mM sodium pyruvate (Merck TMS-005-C), 2.5 g/L glucose (Sigma G8769), and 1x Antibiotic–Antimycotic (Gibco, 15240062) in a humidified 37 °C incubator with 5% $CO_2$. RES-A and RES-B cells additionally received 10 μM enzalutamide (MedChemExpress HY-70002) with each cell splitting/feeding. VCaP cells were cultured in DMEM (Gibco) supplemented with 10% FBS in a humidified 37 °C incubator with 5% $CO_2$.

For experimental treatments, ~$1 \times 10^6$ cells were seeded into 5-cm culture plate dishes, and allowed to settle before exposure to 10 μM enzalutamide or DMSO vehicle control (0.1%) for 48 h or 168 h. The additional LNCaP cells (ATCC) and RES-C cells were cultured in a humidified CO2 incubator at 37 °C in Gibco™ RPMI 1640 (1X) media (Thermo Fisher Scientific) supplemented with 10% FBS (Gibco standard FBS, Thermo Fisher Scientific), 2 mM L-Glutamine (Gibco®, Thermo Fisher Scientific), and a combination of 100 U/ml Penicillin and 100 μg/ml Streptomycin (Gibco® Pen Strep, Thermo Fisher Scientific). The enzalutamide resistant LNCaP RES-C cell line was generated by passaging of LNCaP cells with continuous treatment with 10 μM enzalutamide for nine months and maintained in the same medium as LNCaP except for the supplementation with 10 μM enzalutamide.

**Generation of resistant VCaP subline derivatives and RNA-seq.** The androgen-sensitive VCaP cell line (passage (p.) 15) was a gift from Dr. Tapio Visakorpi, Tampere University, Finland. Cells were cultured in RPMI 1640 supplemented with 10% DCC–FBS, 1% L-glutamine, 1% A/A, and 10 nM testosterone (T) for seven months to establish T-dependent subclone VCaP-T. VCaP-T cells were then cultured at low testosterone (0.1 nM) for 10 months to establish VCaP-CT, an androgen-independent cell line able to grow despite low testosterone. VCaP-CT was then cultured at 10 μM enzalutamide until the cells regained the ability to grow despite enzalutamide, creating enzalutamide resistant cell line VCaP-CT-ET. Another cell line was created by incubating first VCaP-CT cells with bicalutamide and subsequently with enzalutamide upon reaching bicalutamide insensitivity. Ultimately these cells also gained the ability to grow despite enzalutamide, creating the multiresistant cell line VCaP-CT-BR-ER.

RNA sequencing of the VCaP cells was performed with Illumina HiSeq 3000. We sequenced 3 replicates, obtaining an average of 111 million paired-end reads per sample. Reads were aligned using STAR aligner v2.5.4b[52] and Ensembl reference genome GRCh38. Genewise read counts were quantified using featureCounts v1.6.2[53] and Gencode release 28 annotations.

**Single-cell sample preparation and sequencing.** LNCaP parental (treated for 48 h with enzalutamide or DMSO), RES-A, and RES-B cells were harvested with 0.05% Trypsin-EDTA (Sigma T3924). After neutralization with complete medium,

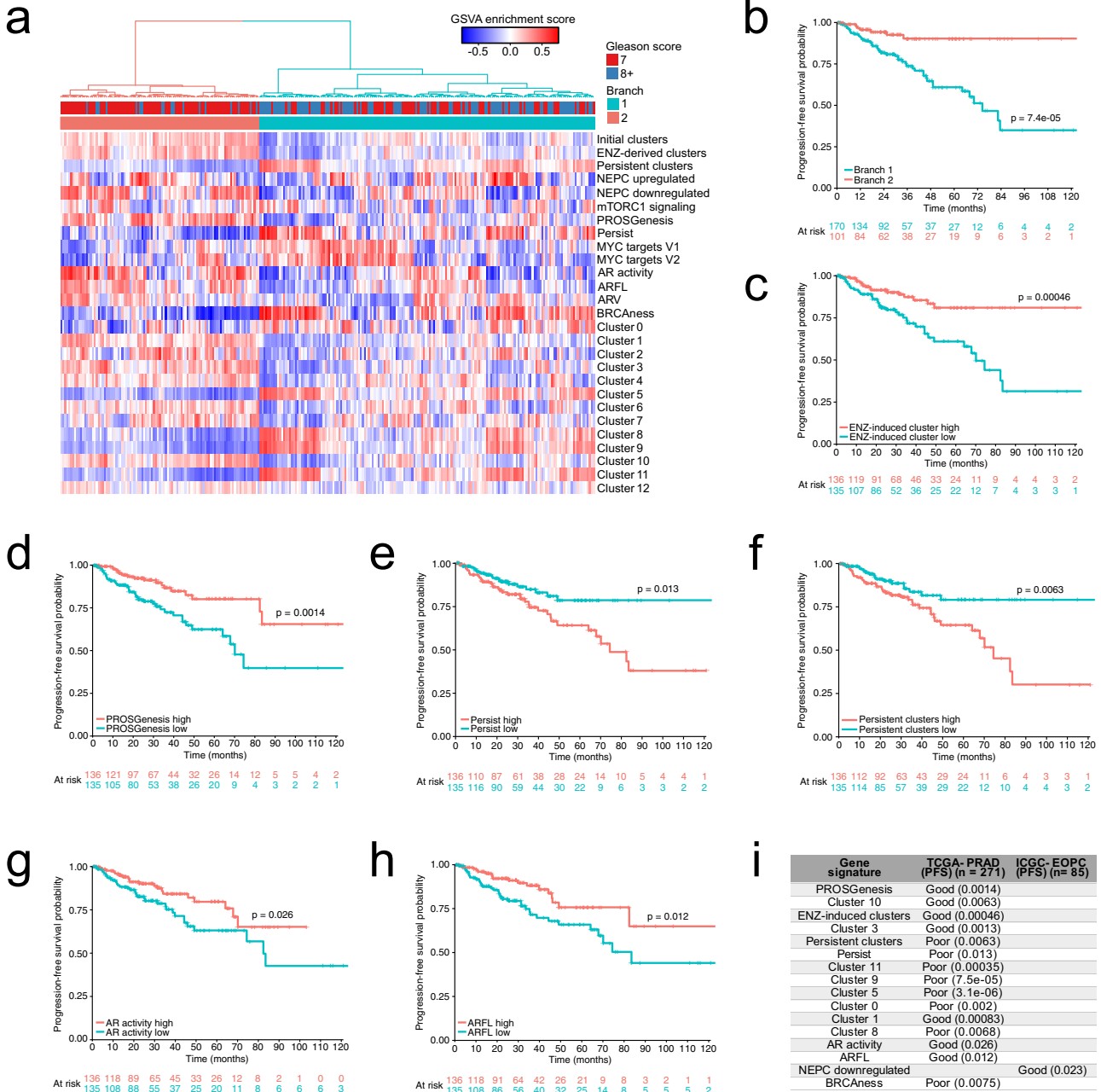

**Fig. 7 Transcriptional signals from persistent prostate cancer cells can be used to stratify untreated patients. a** Heatmap of gene set variation analysis (GSVA) enrichment scores for all single-cell (sc)-derived gene signatures in the TCGA-PRAD cohort, including the marker gene sets for each scRNA-seq cluster. Hierarchical clustering of the GSVA scores was used to separate the samples into two groups, labeled Branch 1 and Branch 2. **b** Kaplan–Meier survival curve for TCGA–PRAD patients stratified into two groups as indicated in panel **a**. The two-sided log-rank *p*-value is shown within the plot. **c–h** Kaplan–Meier survival curves for TCGA–PRAD patients stratified into two groups based on median GSVA score for ENZ-induced cluster, PROSGenesis, Persist, persistent cluster, AR activity, and ARFL gene signatures. In each plot, the two-sided log-rank *p*-value is indicated above the plotted curves. **i** Summary table of gene signature GSVA score associations with progression-free survival (PFS) in the TCGA–PRAD and ICGC–EOPC datasets. Only gene signatures significantly associated with PFS in one or both datasets are shown. Good indicates that a higher score for the signature (a score higher than the median) is associated with better survival outcome, while poor indicates that a higher signature score is associated with worse survival outcome. Two-sided log-rank *p*-values are shown in parentheses. For each dataset, the header indicates the number of samples included. ENZ = enzalutamide, NEPC = neuroendocrine prostate cancer. See also Supplementary Fig. 7.

centrifugation (300 × *g* for 5 min), and resuspension in PBS/0.5% BSA, the cells were filtered through a 35-μm Cell Strainer (Corning 352235) and a single-cell suspension of living cells was acquired through sorting on a FACS Aria II cell sorter. The cell concentration of the single-cell suspension was assessed with a Countess II FL Automated Cell Counter and ~3 × 10⁴ cells were pelleted (300 × *g* for 5 min) for further processing for using the Chromium Single Cell 3′ Library, Gel Bead & Multiplex Kit, and Chip Kit (v3, 10x Genomics).

For the additional LNCaP parental and RES-C cells, 1 million cells were thawed in RPMI (Gibco) with 10% FBS (Gibco) and centrifuged at 300 g for 5 min. The cells were then suspended in PBS with 0.04% BSA (Ambion) and filtered with Flowmi™ cell strainer (Bel-Art). Before loading, the cells' viability and concentration was determined using Trypan blue with Cellometer Mini Automated Cell Counter (Nexcelom Bioscience). Chromium Single Cell 5′ RNA-seq was performed using the 10X Genomics Chromium technology, according to the

Chromium Next GEM Single Cell V(D)J Reagent Kits v1.1 kit User guide CG000208 Rev D with loading concentration of 1000–2000 cells/µl.

The LNCaP–ENZ168, VCaP, and VCaP–ENZ48 single-cell RNA-seq samples were prepared with Drop-seq[54] using the Dolomite cell encapsulation system (Dolomite Bio). Cells were trypsinized with TrypLE™ Express Enzyme (ThermoFisher Scientific, #12604021), spun down (5 min at $300 \times g$), and washed with 0.1% BSA–PBS. After pelleting, the cells were resuspended in plain PBS and passed through a 40-micron filter. The number of viable cells was estimated with the use of trypan blue staining and Fuchs–Rosenthal hemocytometer chamber. The concentration of cells was brought down to $3 \times 10^5$ cells/mL in 0.1% BSA–PBS. For single-cell encapsulation, single-cell suspension, beads in lysis buffer and oil were connected with the loops and tubing to the Mitos P pumps and run through the glass microfluidic chip at the following flow rates: 100 µL/min (Oil channel), 20 µL/min (Bead channel); 350 mbar (Cell channel). Droplets were separated by centrifugation and beads counted with the use of Fuchs–Rosenthal hemocytometer chamber and up to 90000 beads were collected into one tube for reverse transcription reaction, exonuclease treatment, and amplification of cDNA library according to the original protocol[54]. Tagmentation of cDNA was performed with the Nextera XT DNA Library Preparation Kit (Illumina, #FC-131-1024). The PCR product was cleaned up with AMPure XP beads, eluted in 10 µL of $H_2O$, and sequenced using Illumina HiSeq 2500 Rapid run.

For scATAC-seq, cell nuclei were isolated following the 10x Genomics Demonstrated Protocol for Single Cell ATAC Sequencing (CG000169-Rev C). Briefly, the cell suspension was washed once in PBS/0.04% BSA, and $2 \times 10^5$ cells were pelleted ($300 \times g$ for 5 min), resuspended in 100 µl of freshly prepared lysis buffer (10 mM Tris-HCl pH 7.4, 10 mM NaCl, 3 mM $MgCl_2$, 0.1% Tween-20, 0.1% NP40 Substitute, 0.01% Digitonin, and 1% BSA), and incubated on ice for 4 min (LNCaP parental cells), 6 min (RES-A), or 5 min (RES-B). The lysates were diluted with 1 ml of wash buffer (10 mM Tris-HCl, pH 7.4, 10 mM NaCl, 3 mM $MgCl_2$, 0.1% Tween-20, 1% BSA), and the nuclei were pelleted ($500 \times g$ for 5 min) and resuspended in 30 µl of 1× Nuclei Buffer (10× Genomics PN-2000153). Successful preparation of intact, isolated nuclei was confirmed through visual inspection in a phase-contrast microscopy, and nuclei concentration was assessed with a Countess II FL Automated Cell Counter, before proceeding immediately to processing for single cell ATAC sequencing using 10× Chromium, 10× Genomics library preparation and the Chromium Single Cell ATAC Reagent Kits (v1) User Guide (CG000168 Rev D).

Sequencing was performed on the Illumina NextSeq500 instrument at the genomics core facility at the Oslo University Hospital, while sequencing of the additional LNCaP parental and RES-C was performed with Novogene Company Limited, Cambridge, UK's sequencing core facility was used with a PE150 NovaSeq sequencer, aiming at 50000 reads per cell.

For scRNA-seq, sequencing reads were processed into FASTQ format and single-cell feature counts using Cell Ranger v3.0.2[55]. Similarly, Cell Ranger ATAC v1.1.0[56] was used to process sequencing reads from scATAC-seq into FASTQ format and peak-barcode counts. In both cases, we used the Cell Ranger prebuilt GRCh38 reference. The LNCaP–ENZ168, VCaP, and VCaP–ENZ48 Drop-seq samples were preprocessed, aligned, and processed to cell count matrices with the Drop-seq tools v2.3.0 pipeline (as described in https://github.com/broadinstitute/Drop-seq/blob/master/doc/ Drop-seq_Alignment_Cookbook.pdf) using default parameters and with the expectation that each sample contained 1000 cells[54]. The pipeline uses the STAR aligner v2.7.3a[52] and Picard Tools v2.18.22 (http://broadinstitute.github.io/picard/). We utilized the human reference genome version GRCh38, along with Gencode annotations version 33.

**Formaldehyde-assisted isolation of regulatory element (FAIRE) sequencing and analysis**. FAIRE was performed on parental and RES-A cells in biological triplicate according to the standard protocol[57]. Prior to FAIRE-seq, cells were cultured for three days in RPMI medium supplemented with 5% DCC FBS and 10 µM enzalutamide was added only to the resistant cell line. Both sublines were then treated with DMSO (control), DHT (10 nM; Sigma Aldrich), enzalutamide (10 µM, Selleckchem), or a combination of DHT and enzalutamide for 18 h. The DNA fragments isolated by FAIRE were used for library preparation with the Roche KAPA library prep kit according to the manual and sequenced on the Illumina HiSeq 2500 to produce 50-bp single-end reads at the Genomics core (KU Leuven) and aligned using bwa v0.7.8-r455[58] against hg19. Duplicates were marked and realigned using Picard 1.118. Peak calling was performed on the aligned files using MACS2 v2.1.0[59]. MSPC v4.0.2[60] was used to jointly analyze the peaks called in the three replicates from each sample. DiffBind v2.14.0[61] was used to explore peak overlaps and to derive consensus peak sets. Read-distribution analysis around common peak sites, MYC-binding sites, and AR-binding sites was performed by counting the average number of reads across replicates for each sample condition in 100 bp bins extending 1 kb up- and downstream of the sites. The read counts were normalized using the average read counts at the flanks of the distribution (100-bp on both sides). The value at the center (position 0) of the distributions was compared between samples using the $t$-test to assess for differences in chromatin openness at these sites.

**Software and statistical testing**. Software and tools utilized in this study are described in Supplementary Table 2. Analyses were performed using R v3.6.3 or

Python v3.7.0, unless otherwise stated. Statistical testing was performed using R v3.6.3. Statistical tests used are indicated in the text and in figure legends. All statistical tests were two-sided. The Shapiro–Wilk test was used to test for normality.

**Single-cell RNA preprocessing and quality control**. The Cell Ranger output was used as the input to Seurat v3.2.0[24,62] for further analysis of the scRNA-seq samples. For each sample, poor-quality cells were filtered based on the number of detected genes, the total number of molecules detected, and the percentage of reads arising from the mitochondrial genome. Specific filtering thresholds for each sample and the associated quality metrics are shown in Supplementary Data 4. To address the effects of cell cycle heterogeneity in the data, each cell was scored for its expression of genes associated with S or G2/M phases (gene sets provided within Seurat) using the Seurat CellCycleScoring function. The difference between the G2/M- and S-phase scores was regressed out using sctransform[63].

**Single-cell RNA clustering**. The mutual nearest neighbor approach fastMNN[64] was used to integrate the four LNCaP samples using 2000 integration features and account for batch effect. Clustering using default parameters and uniform manifold approximation and projection (UMAP) nonlinear dimensionality reduction were performed using Seurat v3.2.0, and we refer to the result as our integrated clusters. The marker genes of each cluster (genes differentially expressed in each cluster compared with all other clusters) and differentially expressed genes between samples were identified with Seurat using the generalized linear model MAST framework v1.12.0[65], using the number of RNA reads as a latent variable. A gene was considered to be differentially expressed with Bonferroni-corrected $p$-value < 0.01, at least 10% of the cells in the cluster expressing the gene, and an average log-fold change of at least 0.25.

**Single-cell RNA cluster and sample characterization**. We utilized hallmark gene sets from the Molecular Signatures Database (MSigDB) v7.0[66,67] to characterize clusters and samples based on their differentially expressed genes. Gene set variation analysis (GSVA) was performed using the GSVA package v1.34.0 to characterize the average expression profile of each cluster. See the "Bulk RNA-seq and clinical data analysis" section for a more detailed description of the method. To characterize the gene expression changes within each cluster between samples, all genes were ranked based on their average log-fold change. The fgsea package v1.14.0 was then used to perform gene set enrichment analysis for the MSigDB hallmark gene sets using 1000 permutations. The differentiation potential of each cell in each sample was predicted using cytoTRACE v0.3.3[31] using R v4.0.4. The RNA velocities of single cells in the scRNA-seq samples were assessed using scVelo v0.2.2[68]. Loom input files for scVelo were generated from the FASTQ files of each sample using loompy v3.0.0, and the metadata for running scVelo (filtered cell identifiers, UMAP coordinates, and cluster information) were extracted from the integrated Seurat object and integrated with the Loom files.

**Single-cell ATAC preprocessing and quality control**. The output of the Cell Ranger ATAC pipeline was used as the input to Signac package v0.2.5[69] for further analysis of the scATAC-seq samples. For each sample, poor-quality cells were filtered based on the following features: strength of nucleosome-binding pattern, transcription start site enrichment score as defined by ENCODE, total number of fragments in peaks, fraction of fragments in peaks, and percentage of reads in ENCODE-blacklisted genomic regions. Specific filtering thresholds for each sample and the associated quality metrics are shown in Supplementary Data 4. Data normalization and dimensionality reduction was performed using Signac with latent semantic indexing (LSI), consisting of term frequency-inverse document frequency (TF-IDF) normalization and singular-value decomposition (SVD) for dimensionality reduction, using the top 50% of peaks in terms of their variability across the samples. The first LSI component reflected sequencing depth across the samples and was not utilized in downstream analyses.

**Single-cell ATAC clustering and analysis**. Integrated clustering of the scATAC-seq samples was performed with harmony v1.0[70] using LSI embeddings. The resulting harmony-adjusted cell embeddings were used as input in the Signac package for UMAP nonlinear dimensionality reduction and clustering using default parameters and the smart local moving (SLM) algorithm for modularity optimization.

A pseudobulk analysis of changes in chromatin accessibility in the scATAC-seq samples was performed by pooling the reads from all good-quality cells in each sample. Transcription start site enrichments were generated using the TSSEnrichment function in Signac. Chromatin-accessibility tracks based on fragment coverage were generated using the Signac CoveragePlot function. Overrepresentation of Reactome pathways was assessed using ReactomePA v1.30.0[71]. Differentially accessible regions in the clusters were identified using Signac with a logistic regression model that predicts group membership based on each gene and uses a likelihood-ratio test to compare the result to a null model, with the total number of peaks as a latent variable. Regions were considered differentially accessible with Bonferroni-corrected $p$-value < 0.05, at least 10% of the cells showing accessibility in the region, and an average log-fold change of at least 0.25. Differentially accessible regions were annotated with their closest gene

using the Signac ClosestFeature function. Visualization of differentially accessible regions in clusters between samples was generated using R package ggradar v0.2.

**Transcription factor motif enrichment of scATAC-seq and transcriptional output**. Transcription factor motif enrichment was performed using Signac in differentially accessible chromatin regions between sample conditions and between clusters in each sample with R package TFBSTools v1.26.0, R package BSgenome.Hsapiens.UCSC.hg38 v1.4.1, and JASPAR database position-frequency matrices retrieved from the R JASPAR2018 data package v1.1.1. The hypergeometric test was used to test for significant motif enrichments, taking into account sequence characteristics of the chromatin regions (e.g., GC frequency). P-values were adjusted with the Benjamini–Hochberg method and motifs with adjusted p-values less than 0.05 were considered to be enriched. Transcription factors that are known to play a role in PC were filtered based on their expression in the single-cell dataset. Chromatin states in scATAC-seq (as defined by the enriched TFs in differentially open-chromatin regions) were connected to transcriptional outputs in the scRNA-seq by assessing for overlap between the target genes of enriched transcription factors and differentially expressed genes in the scRNA-seq clusters. Transcription factor target genes were obtained using the GTRD database v18.06[72] and selecting those with differentially accessible regions observed between castration-resistant prostate cancer and prostate cancer patients in Uusi-Mäkelä et al[46].

**Integration of scRNA-seq datasets and scRNA- and scATAC-seq datasets using label transfer**. The clusters identified from the integrated clustering of scRNA-seq from LNCaP, LNCaP–ENZ48, RES-A, and RES-B (Fig. 3a) were queried in additional scRNA-seq samples (alternative LNCaP parental, LNCaP–ENZ168, RES-C, VCaP parental, and VCaP–ENZ48) (Fig. 1a) using the label-transfer approach implemented in Seurat v3.2.0[24]. The additional scRNA-seq samples were individually clustered and anchors were identified for each additional scRNA-seq sample (the query) and the LNCaP-integrated clusters (the reference). This was done using the FindTransferAnchors function with principal component analysis (PCA). The anchors were used to transfer cluster-label identifiers between the two data types using the TransferData function. Each cell in the query was assigned the cluster label with the highest prediction score, and only query cells with prediction scores above 0.5 were considered to have been successfully label transferred.

LNCaP, LNCaP–ENZ48, RES-A, and RES-B had scRNA-seq and scATAC-seq data available for each sample (Fig. 1a). These data types were integrated using the cluster-label transfer procedure as implemented in Signac v0.2.5 and Seurat v3.2.0. Each scRNA-seq sample was clustered individually and its cluster labels were projected onto the matching, individually clustered scATAC-seq sample, or vice versa. The clustering resolution of each sample was assessed and decided using clustree v0.4.3[73]. Briefly, RNA-seq expression levels were imputed from the scATAC-seq data by defining for each gene a genomic region, including the gene body and 2 kb upstream of the transcription-start site and taking the sum of scATAC-seq fragments within the region. Anchors were identified for condition-matched scRNA- and scATAC-seq samples using the FindTransferAnchors function and canonical correlation analysis (CCA) was performed on the scRNA expression values and the scATAC-imputed gene expression values. The anchors were used to transfer cluster-label identifiers between the two data types using the TransferData function. Each cell in the query was assigned the cluster label with the highest prediction score, and only query cells with prediction scores above 0.3 were considered to have been successfully label transferred. We tested various prediction-score thresholds and found that approximately 50% or more cells were label transferred between data types in all samples using a threshold of 0.3 (Supplementary Table 3).

**Signature gene selection**. To generate the PROSGenesis signature, we extracted the gene expression profile associated with the regenerative mouse prostate luminal 2 cells reported in Karthaus et al[30] and found 78 genes with homologs in humans that were profiled in our scRNA-seq dataset. The mTORC1 signaling and MYC target gene signatures were obtained from the hallmark gene sets from the Molecular Signatures Database (MSigDB). Other signature gene sets were retrieved from previous publications or from our scRNA-seq data analysis as indicated in the main text.

**Bulk RNA-seq and clinical data analysis**. Each gene signature or set was assessed for enrichment and scored in a sample using the GSVA package v1.34.0, which is a nonparametric, unsupervised method for estimating gene set enrichment of each sample from gene expression data. For GSVA analysis, first, scale normalization at the seventy-fifth percentile based on the DSS package[74] was applied to the raw read counts from samples in datasets where these counts were available. For the TCGA and ICGC cohorts, we then filtered out genes with a zero count in any of the tumor samples. For each gene, GSVA performed a Poisson kernel transformation based on its empirical cumulative density function (CDF) across all samples. For RNA-sequencing datasets where only log-normalized expression values rather than raw counts were available, Gaussian kernels were utilized instead of Poisson kernels in the GSVA calculation. The kernel-transformed expression values were then

converted to ranks for each sample across all genes and the ranks were normalized to centered at zero. Next, for a given gene signature or set, following a similar procedure as GSEA[67], the Kolmogorov–Smirnov-like random-walk statistics were calculated using the normalized ranks based on two statistics: (1) a running sum of the genes that belong to the gene set. It is denoted as $S_1$. (2) A running sum for the genes that do not belong to the gene set. It is denoted as $S_2$. For sample $j$, and gene signature $k$, we define $ES_{jk}^+$ as the largest positive deviations from zero of $S_1-S_2$, and $ES_{jk}^-$ as the smallest negative deviations from zero of $S_1-S_2$. The final GSVA-enrichment score of sample $j$ and gene signature $k$ is $|ES_{jk}^+| - |ES_{jk}^-|$. The GSVA enrichment score emphasizes genes in pathways that are concordantly activated in one direction only, either overexpressed or underexpressed relative to the overall population. For pathways containing genes strongly acting in both directions, the deviations of $|ES_{jk}^+|$ and $|ES_{jk}^-|$ will cancel each other out and show little or no enrichment.

In cases where the expression of a gene set was assessed at the single-cell level, the AddModuleScore function in Seurat was used to generate an average expression score per cell. Survival analyses were performed using the survival package v3.2-3 and Kaplan–Meier curves were plotted using the survminer package v0.4.8. For single-signature survival analyses, median GSVA score was used to stratify patients into low- and high-expressing groups for the signature. For survival analyses of multiple signatures, samples were clustered using their GSVA-enrichment scores for each signature using Euclidean distance and hierarchical clustering. The clustering result was then used to define the two-group split of samples for the survival analysis.

We utilized a published scRNA-seq dataset of PC patient tumor samples from Chen et al.[38] to assess for the presence of our gene signatures in different cell types. The data were processed and visualized according to the code provided as part of the publication (https://github.com/chensujun/scRNA) using Seurat v3.2.0. Cell types were identified from the data using the marker genes reported in Fig. 1B of the publication.

**Spatial transcriptomics analysis of primary prostate cancer tissue**. Two sections of cryopreserved PC tissue were obtained from one patient (pT = 2b, T1c, Gleason 6, PSA 3.5 ng/mL). The use of clinical material was approved by the ethical committee of the Tampere University Hospital. Written informed consent was obtained from the donor. The tissue sections were profiled for spatial transcriptomics using the Visium Spatial library preparation protocol from 10x Genomics with a resolution of 55 μm (1–10 cells) per spot. The tissues were cryosectioned at 10-μm thickness to the Visium library preparation slide, fixed in ice-cold 100% methanol for 30 min, hematoxylin and eosin (H&E) stained with KEDEE KD-RS3 automatic slide stainer, and the whole slide was imaged using Hamamatsu NanoZoomer S60 digital slide scanner.

Sequencing library preparation was performed according to the Visium Spatial Gene Expression user guide (CG000239 Rev D, 10x Genomics) using a 24-min tissue-permeabilization time. Sequencing was done on the Illumina NovaSeq PE150 sequencer at Novogene Company Limited, Cambridge, UK's sequencing core facility, aiming at 50,000 read pairs per tissue-covered spot.

Sequenced data were first processed using Space Ranger v1.2.0 from 10x Genomics to obtain per-spot expression matrices for both sections. Downstream processing and clustering was then performed using Seurat v3.2.0. Normalization of the data was performed with sctransform to account for differences in sequencing depth across spots. Clustering was performed using the FindClusters function with a resolution parameter value of 0.8. The resulting clusters were found to correspond to histological characteristics of the tissue. The AddModuleScore function of Seurat was used to score the spots for our scRNA-seq-derived gene signatures, as well as length-matched random housekeeping gene signatures from the Housekeeping and Reference Transcript Atlas v1.0[75]. The distributions of the gene expression scores for the housekeeping gene sets and our scRNA-seq signatures were compared to determine the 90th percentile as a score cutoff at which we considered a spot to have high expression of the scRNA-seq signature, allowing for 5% false positives (spots scoring above the threshold for housekeeping gene sets).

To validate our spatial transcriptomics findings, we utilized prostate sections 1.2, 2.4, and 3.3 from the spatial transcriptomics publication by Berglund et al.[39]. H&E images and spot count matrices were provided by the Lundeberg lab. Processing, clustering, and signature scoring of the data was performed identically to sections of prostate A and B, but requiring that each spot would have a minimum of 500 read counts. Similar to the analysis for prostate A and B, the 90th percentile cutoff for high- versus low-scoring spots for the gene set enrichment was assessed and confirmed using comparisons to housekeeping gene set scores for each spot.

**Reporting summary**. Further information on research design is available in the Nature Research Reporting Summary linked to this article.

## Data availability

The single-cell RNA, single-cell ATAC, FAIRE-seq, and RNA-seq data generated in this study have been deposited in the Gene Expression Omnibus (GEO) archive under accessions GSE168669 and GSE168733. The spatial transcriptomics data are available at the

European Genome-Phenome Archive (EGA) under identifier EGAS00001000526. Other publicly available datasets utilized but not generated in this study were AR- and c-MYC-binding site maps (GEO archive: GSE73994), bulk RNA sequencing of LNCaP samples analyzed using single-cell methods in this study (GEO archive: GSE130534), xenografts of AR-positive/NE-negative and AR-negative/NE-positive CRPC tumors (GEO archive: GSE124704 and GSE126078), single-cell RNA-seq of 13 treatment-naive prostate tumor samples (GEO archive: GSE141445), LNCaP xenograft models of CRPC (Supplementary File 1 in King et al., 2017 [https://doi.org/10.18632/oncotarget.22560]), patient RNA sequencing from enzalutamide responders and nonresponders [https://doi.org/10.1073/pnas.1922207117], RNA sequencing from SU2C CRPC patient samples [https://doi.org/10.1073/pnas.1902651115], RNA sequencing from SU2C West Coast DT patient samples [https://doi.org/10.1016/j.cell.2018.06.039], spatial transcriptomics data from prostate tissue sections 1.2, 2.4, and 3.3 [https://doi.org/10.1038/s41467-018-04724-5], TCGA-PRAD RNA-seq [https://portal.gdc.cancer.gov/], and ICGC-EOPC RNA-seq [https://doi.org/10.1016/j.ccell.2018.10.016]. Databases utilized in the study were the Molecular Signatures Database (MSigDB) v7.0 (http://www.gsea-msigdb.org/gsea/msigdb/index.jsp), the GTRD database v18.06 (https://gtrd.biouml.org/), and the Housekeeping and Reference Transcript Atlas v1.0 (http://www.housekeeping.unicamp.br/). The remaining data are available within the article, Supplementary Information, or Source Data file. Unique biological materials are available from the corresponding authors upon reasonable request. Source data are provided with this paper.

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

## Acknowledgements

We thank the Genomics Core Facility at Institute for Cancer Research (OUH; Oslo University Hospital, Oslo, Norway) for the support during preparation of the scATAC- and RNA-seq, the Tampere University histology core facility and Sari Toivola for skillful assistance during Visium experiments, and the UEF Bioinformatics Center, University of Eastern Finland, Finland. The study was financially supported by the Finnish Cultural Foundation (ST), Academy of Finland (#312043 (MN), #310829 (MN), #324009 (KK), #328928 (KK), #333545 (KG, EMV, and SH), #3121330724 (TM)), Finnish Cancer Society #3122800563 (TM) Cancer Foundation Finland (MN, KK, and KG), Sigrid Jusélius Foundation (MN, KK, and KG), Emil Aaltonen Foundation (KG), Finnish Cancer Institute (MN), Norwegian Cancer Society (#198016-2018)(AU, NE), Competitive State Research Financing of the Expert Responsibility area of Tampere University Hospital (MN, TLJT, TV, and TM), The Norman Jaffe Professorship in Pediatrics Endowment Fund (SC), MD Anderson Colorectal Cancer Moon Shot Program (SC), Oncode Institute (SP), Finnish Cultural Foundation North Savo Regional fund (KK, RK), University of Eastern Finland Doctoral Programme in Molecular Medicine (RK), K. Albin Johansson Foundation (KK), John Black Foundation (IM), Human Cell Atlas Seed Network-Retina (WW), Chan Zuckerberg Institute (WW), NIH R01CA183793 (WW, SC), NIH R01CA239342 (WW), NIH R01CA158113 (WW), P30CA016672 (WW), the Fonds Wetenschappelijk Onderzoek-Vlaanderen #GOA9816N (FC), KU Leuven #C14/19/100 (FC), Kom op tegen Kanker #KOTK (FC, WD), Cancer Research UK # A22744 (GA, DW, KN, and PC), and Cancer Research UK #C57899/A25812 (AE, ADL). The results published here are in part based on data generated by The Cancer Genome Atlas project established by the NCI and NHGRI.

## Author contributions

A.U. and M.N. designed and supervised the research. A.U. and S.T. wrote the initial paper draft. S.T. performed the formal analysis and prepared the figures. R.K. and H.K. generated LNCaP RES-C cells. N.E., R.K., M.L., K.N. and D.W. prepared the single cell sequencing. S.C. performed the data analysis of the TCGA and ICGC bulk RNA-seq PC cohorts. F.H. and P.C. contributed to the single-cell data analysis. K.J.G. supervised the spatial transcriptomics. S.H. and E.M.V. performed the spatial transcriptomics experimental work. A.K., A.E. and T.H. performed the spatial transcriptomics data analysis. S.P. performed the FAIRE sequencing. T.T. evaluated histology and immunohistochemistry. H.S. and M.B. generated the VCaP cell lines derivatives resistant to treatments. R.N. contributed the VCaP cell line bulk RNA-seq preprocessing analysis. M.N., A.U., F.C., T.V., T.L.J.T., K.J.G., T.M., K.K., A.D.L., G.A. and W.W. provided material and resources. A.U., S.T., M.N., K.J.G., W.D., K.K., F.C., I.G.M., A.D.L., G.A. and W.W. performed paper draft editing. All authors reviewed and edited the final paper.

## Competing interests

GA receives a reward from the Institute of Cancer Research for his role as an inventor of abiraterone. GA has received honoraria, consulting fees, or travel support from Janssen, Astellas, Pfizer, Novartis, Bayer, Amgen, AstraZeneca, Sanofi, and Sapience, grant support from Janssen and Astellas, and is a principal investigator for clinical trials sponsored by Janssen, Pfizer, and Astellas. TM receives consultant fees from Astellas, Janssen, and Bayer; lecture fees from Novartis, Janssen, and Sanofi. He is a stockholder of Arocell ab. The remaining authors declare no competing interests.
