## [Peer Review File · Nature Communications]

Single-cell ATAC and RNA sequencing reveal pre-existing and persistent cells associated with prostate cancer relapseReviewers' Comments:

Reviewer #1:

Remarks to the Author:

The manuscript entitled "Single-cell ATAC and RNA sequencing reveal pre-existing and persistent subpopulations of cells associated with relapse of prostate cancer" provides a tour-de-force molecular account as to the subpopulations of prostate cancer cells that maybe potentially drive the emergence of castration-resistant-prostate cancer and lethal disease. The investigators/authors were able to accomplish using innovative state-of-the art technology as single-cell ATAC to identify genes responsible for the persistent subpopulations. This question of pre-existing populations of prostate cancer cells responsible for therapeutic resistance has been identified over many years, but it is only in the present study that the authors addressed it with mechanistic insight and delivered novel findings as to the molecular landscape. The results are innovative and mechanistically robust and the study is executed with a tremendous precision as to the translational relevance of this analytical profiling to human disease of lethal prostate cancer.

I only have a minor comment regarding this strong and innovative manuscript. It needs to become more succinct. The length of the manuscript can be shortened, specifically in the "Materials and Methods" (detailed technical descriptions) and the "Results" section (some redundant data description).

Reviewer #2:

Remarks to the Author:

The stated goal of the manuscript entitled 'Single-cell ATAC and RNA sequencing reveal pre-existing and persistent subpopulations of cells associated with relapse of prostate cancer' by Taavitsainen et al was to 'identify molecular predictors of therapy response based on the presence of treatment-persistent and pre-existing cells'. They show that subpopulations of treatment-persistent cells with stem-like and regenerative properties foster alternative trajectories of enzalutamide resistance in prostate cancer. Alternative transcriptional patterns of resistance are induced by divergent chromatin reprogramming. Transcriptional enrichment of signals from treatment-persistent cells in culture can be applied to stratify patient outcomes in both early stage treatment-naive and treatment-exposed tumors. The paper is expertly written and links observations in culture to patient outcomes. The cluster-specific signatures generated here have to chance to change clinical practice by identifying existing subclones with metastatic potential in primary untreated tumors. The data are robust and expertly analyzed. The interpretations are logical and the figures are understandable.

A lingering question will be how to differentiate benign basal epithelia from tumor epithelia with metastatic potential based on the high correlation of these signatures with benign epithelia. Perhaps some combined score will be necessary. Regardless, these data lead to new insights about the heterogeneity and clinical relevance of cell culture models and will hopefully lead to new insights about the preexistence of prostate cancer clones with high metastatic potential.

Reviewer #3:

Remarks to the Author:

In the manuscript "Single-cell ATAC and RNA sequencing reveal pre-existing and persistent subpopulations of cells associated with relapse of prostate cancer" Taavitsainen et al. employed single-cell ATAC-seq and RNA-seq in cell culture models of treatment response and resistance to enzalutamide to identify pre-treatment biomarkers in prostate cancer. The authors identify prostate cancer cell populations with stem-like features before treatment that persisted after treatment and show changes in chromatin associated with enzalutamide treatment and resistance. They indicate that these stem-like cells might be used to predict response to treatment and finally describe that their single cell data on highlights molecular predictors of treatment outcome.

My criticisms are below:

1) It looks that single cell RNA-seq and scATAC-seq were performed once for each sample. How can the authors rule out that e.g. sample/treatment specific clusters in scATAC are not driven by technical variations rather than biology? The integration analysis shows some correlation between scATAC-seq and scRNA-seq clusters. Clusters 6 and 7 from scATAC (specific to ENZ resistant clones) don't have a corresponding cluster in RNA (Fig 3E) and on the other hand cluster 2/3 from RNA which are ENZ-induced correlate best with cluster 2 from scATAC which is considered persistent cluster. This discrepancy could be caused by challenges using computational integration of the clusters, batch differences in the samples used for scATAC and scRNA-seq. To answer this question, it would be good to add independent replicates for each modality or generate a dataset with recently available kits that allow profiling of both modality from the same nucleus (Multiome). I acknowledge that at this time it might be very difficult to generate additional data, but think it would be important to have further support to strengthen the findings from scATAC-seq.

2) For all samples (scRNA and scATAC) it would be important to report crucial quality control metrics for single cell experiments such as genes and UMIs/cell detected passing quality control, fraction of reads in cells for RNA and for fragments/cell number of cells for ATAC. This would also help to evaluate the statement of the authors that "filtering criterium were adjusted per sample to preserve a maximal number of cells", which potentially can introduce bias. At the minimum these criteria need to be mentioned alongside statistic how many cells were filtered since this might potentially introduce significant bias for downstream analysis. I am also wondering why the authors choose different permeabilization times for scATAC for different samples and wonder if that might lead to observed changes in lower overall TSS enrichment (a measure of signal-to-noise ratio), e.g. I would not expect all genes to have lower TSS enrichment in one sample due to treatment. Here it would be important to show more detailed analysis, for example is it a specific group of genes that has lower TSS enrichment? Genome browser tracks for representative loci for example constant ones and variable ones would be helpful as well.

3) Cluster 11 is described both as G2/M-phase related and stem-like. It is not very clear how distinction was made here since most of the highly expressed genes are general cell cycle markers. Are there specific markers that distinguish proliferating cells (which also go through G2/M) from stem-like cells? In 4F/G authors show RNA velocity and describe it as differentiation trajectory, based on this analysis the differentiation trajectories seem to be different between samples and for example clusters 8/9 in RES-A seem to be less differentiated compared to cells in the same sample in cluster 11 (stem-like). How do the authors reconcile this?

NCOMMS-21-12818: Point-by-point response to the reviewers' comments

We thank the reviewers for their assessment of our manuscript and their constructive comments and suggestions. Below is a point-by-point response and notations for changes/updates to the manuscript results.

Reviewer #1, expert in prostate cancer therapy and resistance (Remarks to the Author):

The manuscript entitled "Single-cell ATAC and RNA sequencing reveal pre-existing and persistent subpopulations of cells associated with relapse of prostate cancer" provides a tour-de-force molecular account as to the subpopulations of prostate cancer cells that maybe potentially drive the emergence of castration-resistant-prostate cancer and lethal disease. The investigators/authors were able to accomplish using innovative state-of-the art technology as single-cell ATAC to identify genes responsible for the persistent subpopulations. This question of pre-existing populations of prostate cancer cells responsible for therapeutic resistance has been identified over many years, but it is only in the present study that the authors addressed it with mechanistic insight and delivered novel findings as to the molecular landscape. The results are innovative and mechanistically robust and the study is executed with a tremendous precision as to the translational relevance of this analytical profiling to human disease of lethal prostate cancer. I only have a minor comment regarding this strong and innovative manuscript. It needs to become more succinct. The length of the manuscript can be shortened, specifically in the "Materials and Methods" (detailed technical descriptions) and the "Results" section (some redundant data description).

Response: We thank the reviewer for the feedback on the manuscript. We have shortened the manuscript text by moving technical descriptions from the Materials and Methods into a Supplementary Information file which contains Supplementary Methods, Supplementary Figures, and Supplementary References.

We partially moved the following:

- Cell lines maintenance and experimental treatment procedures
- Generation of resistant VCaP subline derivatives
- Single cell preparation technical details
- some experimental information regarding FAIRE assay
- Technical description of "Single-cell RNA clustering" and "Cluster and sample characterization"
- Transcription factor motif enrichment analysis and transcriptional output
- Technical description of the use of GSVA in "Bulk RNA-seq and clinical data analysis"
- Technical information on the Spatial transcriptomics analysis

We also edited data descriptions in the Results to be more succinct where possible.

For example, we edited the following:

- Characterization of the previously published bulk RNA-seq samples from LNCaP using gene signatures other than Persist, PROSGenesis, and the NEPC signatures.
- Description of the results from the FAIRE-seq analysis, which was used to confirm findings from the scATAC-seq (description of Supplementary Fig. 1 and 2).

Reviewer #2, expert in prostate cancer genomics and scRNA-seq (Remarks to the Author):

The stated goal of the manuscript entitled ‘Single-cell ATAC and RNA sequencing reveal pre-existing and persistent subpopulations of cells associated with relapse of prostate cancer’ by Taavitsainen et al was to ‘identify molecular predictors of therapy response based on the presence of treatment-persistent and pre-existing cells’. They show that subpopulations of treatment-persistent cells with stem-like and regenerative properties foster alternative trajectories of enzalutamide resistance in prostate cancer. Alternative transcriptional patterns of resistance are induced by divergent chromatin reprogramming. Transcriptional enrichment of signals from treatment-persistent cells in culture can be applied to stratify patient outcomes in both early stage treatment-naive and treatment-exposed tumors. The paper is expertly written and links observations in culture to patient outcomes. The cluster-specific signatures generated here have the chance to change clinical practice by identifying existing subclones with metastatic potential in primary untreated tumors. The data are robust and expertly analyzed. The interpretations are logical and the figures are understandable.

A lingering question will be how to differentiate benign basal epithelia from tumor epithelia with metastatic potential based on the high correlation of these signatures with benign epithelia. Perhaps some combined score will be necessary. Regardless, these data lead to new insights about the heterogeneity and clinical relevance of cell culture models and will hopefully lead to new insights about the preexistence of prostate cancer clones with high metastatic potential.

Response: We thank the reviewer for this feedback on our work. We agree that the ability to differentiate between benign basal epithelia and tumor epithelia with metastatic potential is indeed a major question in the prostate cancer field, if not in the whole field of cancer research, and this will warrant many future studies. Recent advances in spatial transcriptomics are facilitating the distinction at a clonal level, in situ, between true benign epithelia, and those benign epithelia that have acquired genomic variants pending transformation to potentially lethal tumor. Some of the co-authors here are engaged in a body of work that will begin to answer this question. Additionally, some recent studies such as those by Chen et al. [PMID: 33420488] and Couturier et al. [PMID: 32641768] have shown the use of copy number alterations in distinguishing between normal and cancer cells, as well as in identifying cells with possible metastatic potential. Further, perhaps a combined score as suggested by the reviewer of copy number alterations and scores for transcriptional signatures such as those presented in our study could be used to nominate cells with treatment resistance and metastatic potential. While exploring this further is out of the scope of our current study, we have added the following to the manuscript discussion to highlight the importance of this question (page 12, lines 39-42):

“As differentiating between benign basal epithelia and tumor epithelia remains one of the major questions of the prostate cancer field, understanding the presence and function of these persistent and potentially regenerative cells in histopathologically non-cancerous regions warrants further studies.”

Reviewer #3, expert in scATAC-seq (Remarks to the Author):

In the manuscript “Single-cell ATAC and RNA sequencing reveal pre-existing and persistent subpopulations of cells associated with relapse of prostate cancer” Taavitsainen et al. employed single-cell ATAC-seq and RNA-seq in cell culture models of treatment response and resistance to enzalutamide to identify pre-treatment biomarkers in prostate cancer. The authors identify prostate cancer cell populations with stem-like features before treatment that persisted after treatment and show changes in chromatin associated with enzalutamide treatment and resistance. They indicate that these stem-like cells might be used to predict response to treatment and finally describe that their single cell data on highlights molecular predictors of treatment outcome.

My criticisms are below:

Reviewer 3 comment:

1) It looks that single cell RNA-seq and scATAC-seq were performed once for each sample. How can the authors rule out that e.g. sample/treatment specific clusters in scATAC are not driven by technical variations rather than biology? The integration analysis shows some correlation between scATAC-seq and scRNA-seq clusters. Clusters 6 and 7 from scATAC (specific to ENZ resistant clones) don't have a corresponding cluster in RNA (Fig 3E) and on the other hand cluster 2/3 from RNA which are ENZ-induced correlate best with cluster 2 from scATAC which is considered persistent cluster. This discrepancy could be caused by challenges using computational integration of the clusters, batch differences in the samples used for scATAC and scRNA-seq. To answer this question, it would be good to add independent replicates for each modality or generate a dataset with recently available kits that allow profiling of both modality from the same nucleus (Multiome). I acknowledge that at this time it might be very difficult to generate additional data, but think it would be important to have further support to strengthen the findings from scATAC-seq.

Response: We thank the reviewer for these comments and valuable feedback. Without replicates, we agree that the question of technical variability is indeed a pertinent one. We now acknowledge this limitation in the discussion as follows (page 13, lines 11-17):

“While we used separate assays for scATAC-seq and scRNA-seq in different cells, we were able to integrate these data using the label transfer method across datasets. These analyses revealed that subpopulations of cells with different chromatin states may lead to multiple transcriptional configurations, including those of persistent cells. Availability of chromatin accessibility and gene expression data from the same cell would further reduce technical variation and enable more in depth characterization of these configurations.”

We have however taken several steps to determine and show that our results on matched single-cell clusters between pairs of scRNA-seq and scATAC-seq data are not due to technical variation as detailed below.

In the laboratory during sample preparation, we minimized the possibility of incurring batch effects by harvesting the cells used for the scRNA- and scATAC-seq experiments from the same wells on the same day and dividing them into equal

aliquots for each experiment. By doing this, we have reduced the potential technical variability across cells sequenced with the two platforms.

To determine that our cluster-matching results are not due to technical variations and to address whether we could assign more cells between the scATAC- and scRNA-seq clusters, we also performed additional analyses. The cluster label transfer process assigns a prediction score for each cell's membership in a cluster from the reference data type (in our case scRNA-seq). The cluster that has the maximal score is the one that the cell most likely belongs to. We required the maximum score to be at least 0.4 out of 1 to provide the most confident label transfer assignments. If a given cell did not have a prediction score of 0.4 or higher for any of the clusters, the cell was not assigned in this analysis. The threshold of 0.4 has been previously used in studies that used the label transfer method to assign cells from scATAC- to scRNA-seq from blood samples [PMID: 33352111]. Given that our samples are from cell lines, we could not identify cell types a priori or use selected chromatin features in the analysis. However, we could use cluster marker genes in the label transfer process to verify that these improve the prediction scores of the label transfer cluster assignments compared to using the most variable genes in the dataset (Figure 1).

Figure 1. Number of scATAC cells distributed by label transfer prediction scores when using the 2000 most variable genes markers (left column) or the cluster marker genes (right column) in each sample, using scRNAseq data as a reference dataset.

We have repeated the analysis in Fig. 3e and Supplementary Fig. 3e using the most variable genes but releasing the label transfer prediction score threshold parameter. We tested various label transfer threshold parameters and have added a new Supplementary Table 6 to show the percentages of scATAC-seq cells that we are able to label transfer to a cluster from scRNA-seq with varying prediction score thresholds. We refer to this table in the Methods. We found that using a threshold of 0.2, more than 90% of the cells get assigned. This high percentage of cells assigned is consistent with the results obtained by Cao et al. [PMID: 30166440] using a Multiome approach in co-assayed cell lines and in which approximately 90% were found to have a corresponding cluster in both RNA and ATAC data.

Based on the above, in Fig. 3e and Supplementary Fig. 3e, we now report the results using 0.3 as a prediction score threshold, in which at least 50% of cells are label transferred between data types. Moreover, we report the proportions of cells assigned from the scATAC-seq to scRNA-seq in the heatmaps in Fig. 3e and Supplementary Fig. 3e.

We have modified the manuscript text to indicate that we find the ENZ-induced scRNA-seq clusters to label transfer in highest proportion to cells in the scATAC-seq samples. Thanks to the suggestion of the reviewer, this new analysis now raises the possibility of scRNA-seq cluster 3 representing a common genomic configuration for ENZ resistance, as this cluster not only label transfers between data types, but also to other ENZ-treated LNCaP cell line samples (RES-C and LNCaP treated with ENZ for 168 hours). We have now added a comment related to these results in the main text (page 6, lines 36-42):

*“The presence of ENZ-induced clusters was confirmed in RES-C (17% of cells in cluster 3) and LNCaP-ENZ168 (79% in cluster 3), suggesting that one week of ENZ treatment is sufficient to give rise to this cluster prior to the development of resistance (**Supplementary Fig. 3c**). As cells from most scATAC-seq clusters were additionally found to correspond to cluster 3 cells from the scRNA-seq (**Fig. 3e**), this suggests that the cells of this cluster may represent a common genomic configuration for ENZ resistance or its development.”*

To show that label transfer could be used as an unbiased method for retrieving correspondences in the two data types in cell lines as opposed to tissue samples, we downloaded scRNA-seq and scATAC-seq from HAP1 cells [PMID: 33821571] and performed label transfer as in our LNCaP samples. We could assign 97.1% of scATAC-seq cells to a scRNA-seq cluster with a prediction score threshold of 0.3, and similarly to the label transfer process in our samples, the number of transferred cells decreases dramatically when the threshold of the score is increased (Figure 2). While we acknowledge that in this case the label transfer performs slightly better, the trend is substantially the same and the number of assigned cells decreases with increased confidence threshold. This suggests a typical behavior of the label transfer method using two types of assays (scRNA-seq and scATAC-seq) in cell lines as compared to label transfer performed using pre-established markers to define cell types in tissue samples.

Figure 2. Label transfer between scATAC-seq and scRNA-seq data in HAP-1 cells. (A) Number of scATAC cells distributed by label transfer prediction scores when using the 2000 most variable genes markers. (B) UMAP visualization of labelled cells clustered in the scRNA-seq and (C) transferred to scATAC-seq. (D) UMAP visualization of scATAC-seq clusters.

While this analysis points at the limitation of the assay used and we agree that performing Multiome would possibly be the only way to further increase the assignment of the cells from one data type to another, we do show consistently high performance of the label transfer method in cell lines.

Reviewer 3 comment:

2) For all samples (scRNA and scATAC) it would be important to report crucial quality control metrics for single cell experiments such as genes and UMIs/cell detected passing quality control, fraction of reads in cells for RNA and for fragments/cell number of cells for ATAC. This would also help to evaluate the statement of the authors that “filtering criterium were adjusted per sample to preserve a maximal number of cells”, which potentially can introduce bias. At the minimum these criteria need to be mentioned alongside statistic how many cells were filtered since this might potentially introduce significant bias for downstream analysis.

Response: We thank the reviewer for bringing this to our attention. We have now added a new Supplementary Table 5 to the revised manuscript that contains the quality control filtering thresholds and criteria for each scRNA-seq and scATAC-seq sample, the sequencing quality metrics of the cells in each sample as reported by 10x Genomics Cell Ranger or Cell Ranger ATAC, and the quality control metrics of the cells in the samples after quality control filtering. We refer to this Supplementary Table 5 within the Methods.

Reviewer 3 comment:

I am also wondering why the authors choose different permeabilization times for scATAC for different samples and wonder if that might lead to observed changes in lower overall TSS enrichment (a measure of signal-to-noise ratio), e.g. I would not expect all genes to have lower TSS enrichment in one sample due to treatment. Here it would be important to show more detailed analysis, for example is it a specific group of genes that has lower TSS enrichment? Genome browser tracks for representative loci for example constant ones and variable ones would be helpful as well.

Response: We thank the reviewer for giving us the opportunity to clarify about the permeabilization time and the suggestion of performing more detailed analyses. Although RES-A and RES-B are both derivatives of LNCaP cells, they are adapted to live in different conditions and have therefore acquired different intrinsic features. In other words, they can be considered different cell lines. For instance, RES-B cells tend to form clumps of cells more than parental LNCaP, hence possibly revealing some changes related to membrane composition. The reason for using different permeabilization times during scATAC-seq sample preparation relates to the fact that the different cell lines react differently to the lysis buffer. Therefore, we carefully optimized the timing for lysis of the cells during nuclear isolation by visual inspection of the quality of the nuclei. This ensured optimal cell lysis while preserving the integrity of the nuclei in a similar manner across the three cell lines.

Based on the fact that these cell lines are indeed three different cell lines cultured in different conditions, only the parental LNCaP are “treated” with the androgen receptor (AR) inhibitor enzalutamide, while RES-A and RES-B are “maintained” in enzalutamide (ENZ). In Urbanucci et al. Cell Reports 2017 [PMID: 28591577], we showed that differential AR signalling in prostate cancer cells is accompanied by changes in chromatin organization that also involves the TSS. We observe similar changes in Fig. 1b, but also in Fig. 2a-b, as well as in Supplementary Fig. 1d and Supplementary Fig. 1f when using FAIRE-seq (a bulk method). Of note, the differences between samples can be appreciated at shared sites (Supplementary Fig. 1d and Supplementary Fig. 1f) and are indeed affected by both treatment with ENZ (for ENZ sensitive cells) and “maintenance” in ENZ (in ENZ resistant cells).

To address the reviewer's comment with more detailed analysis, we have assessed the enrichment scores around the TSS using different categories of genes such as housekeeping genes and gene sets defined in the Molecular Signatures Database (MSigDB) collection: androgen response genes, MYC target genes, and genes belonging to the mTORC signaling pathway. We now report the scATAC-seq signal at the TSS of these gene sets in the different samples/cell lines in Figure 3 below, and have modified Fig. 1b to include 3 additional panels of this TSS enrichment analysis around AR target genes, housekeeping genes, and MYC target genes. This analysis reveals that the samples from parental lines (treated with DMSO or ENZ) have similar TSS enrichment patterns irrespective of the gene set, as do the ENZ resistant samples. Consistent with our initial analysis and our previous findings (Urbanucci et al., 2017), we observe lower enrichment of the scATAC-seq signal around the TSS for all these gene sets in the ENZ resistant samples

Figure 3. Mean enrichment score of scATAC-seq reads around TSS of (A) housekeeping, (B) mTORC signaling pathway, (C) androgen response, and (D) MYC targets genes. Sample comparisons are indicated using colored dots within the plots and the Wilcoxon rank sum p-value is shown with asterisks (* p-value < 0.05, ** p-value < 0.01, * p-value < 0.001).**

As suggested by the reviewer, we now also report representative gene loci related to this detailed analysis above in Supplementary Fig. 1a. As an example, the *KLK2/3* gene locus displays depletion of the scATAC-seq reads particularly at the TSS regions of both these AR target genes, with additional peaks found outside the TSS in the ENZ resistant lines. However, we also find that for some gene loci, exemplified by the locus of the housekeeping gene *GAPDH*, there seems to be no variation in scATAC-seq signal in any of the lines.

These data show that the effect observed in Fig. 1b is due to biology and not a technical artefact. We understand that the signal to noise at the TSS is also used as a metric for assessing the quality of ATAC-seq data. Therefore, we now report detailed sample quality metrics in Supplementary Table 5.

Reviewer 3 comment:

3) Cluster 11 is described both as G2/M-phase related and stem-like. It is not very clear how distinction was made here since most of the highly expressed genes are general cell cycle markers. Are there specific markers that distinguish proliferating cells (which also go through G2/M) from stem-like cells?

Response: We thank the reviewer for this question and the opportunity to clarify.

We understand that the use of “stem-like” cells as a term can create some confusion given the fact that this term is used with several meanings in different contexts. We have therefore renamed the Stem-Like signature as the “Persist” signature to highlight their most important feature, which is the persistence of these cells during enzalutamide treatment. We find these cells to be characterized by proliferative markers as well as other markers potentially indicative of stem-like or regenerative capability as detailed below. In another manuscript by some of the authors, currently under revision with this journal (NCOMMS-21-15868-T), we observe persistent treatment resistant clones in biopsies taken before and after androgen deprivation therapy which have undergone spatial transcriptomic interrogation with consequent retention of spatial architecture. In the future this could be a useful validation dataset to anchor the “Persist” signature in a spatial context. However, we describe below our analytical efforts to validate the distinction between “stem-like” and “proliferating” signatures.

We define stem cells as rapidly cycling cells with short G1 and G2 phases as compared to M and S phases [PMID: 28985214, PMID: 22084091, PMID: 26876348], by definition these cells are characterized by cell cycle genes especially related to DNA synthesis (e.g. *TOP2A*, *POLQ*, etc). As stem-like cells predominantly synthesize DNA, they are more prone than other cells to incur in DNA damage when actively replicating [PMID: 28475867, PMID: 26876348]. In our study, interestingly, for example, we do observe that DNA repair genes are indeed upregulated selectively in cluster 11. We have already partially reported this characterization of cluster 11 beyond cell cycle markers in Fig. 4b. We had generated a list of genes expressed more highly in Cluster 11 compared to all other clusters and found that the cluster was “*characterized not only by cell cycle genes, but also by the expression of genes involved in chromatin remodeling and organization (CTCF, LAMINB, ATAD2), increased cell cycle turnover and stemness (FOXM1), and DNA repair (BRCA2, FANCI, RAD51C, POLQ).*”

Thanks to the opportunity given to us by the reviewer, we looked back at the genes expressed more highly in Cluster 11 and noted that cluster 11 is dominated by high expression of targets of two transcription factors associated with stemness: FOXM1 and HES6. Our groups have previously characterized these two transcription factors as driving specific CRPC subtypes [PMID: 28899970; PMID: 24737870; PMID: 25006183]. FOXM1 target genes (*AURKB*, *AURKA*, *BIRC5*, *PLK1*, *CCNB1*, *CCNA2*, *GTSE1*, *CDK1*, *CDKN3*, and *CENPA*) were reported in Ketola et al., 2017 [PMID: 28899970] and are highly expressed in cluster 11 (Figure 4 below), and targets of HES6 according to Ramos-Montoya et al., 2014 [PMID: 24737870] such as *PLK1* are also cluster 11 marker genes. We now systematically queried HES6 target genes identified in Ramos-Montoya et al., 2014 [PMID: 24737870] and found that a HES6 target footprint is highly notable in cluster 11 (Figure 5 below).

Figure 4. Gene expression dot plot of FOXM1 target genes identified in Ketola et al. [PMID: 28899970].

Figure 5. Gene expression dot plot of a core gene set of HES6 target genes identified in Ramos-Montoya et al. [PMID: 24737870].

To further tease out stem-like markers from markers of proliferation, we also queried the expression levels of a set of 335 genes reported in a core embryonic stem cell-like module based on human and mouse data [PMID: 18397753] in our scRNA-seq clusters. We identified 34 genes with functions not limited to the cell cycle that had the highest expression in Cluster 11 (Figure 6 below). We then performed a GeneMania (Genemania.org) analysis of these genes and found that the 34 genes enriched for functions such as signal transduction in response to DNA damage (6 genes, FDR 2.59e-4), apoptotic nuclear changes (3 genes, FDR 6.73e-3), and signal transduction by p53 class mediator (6 genes, FDR 1.01e-2). This analysis therefore also points at increased expression of genes involved in DNA damage response as a potential key feature of cluster 11. Interestingly, 10 genes from the stem-cell module were already present in the dot plot in Fig. 4b (*TOP2A*, *AURKA*, *AURKB*, *CCNA2*, *BUB1*, *BUB1B*, *LMNB1*, *WEE1*, *GMNN*, and *EIF4EBP1*).

We have added 6 genes from this set of 34 to the dot plot in Fig. 4b to further characterize Cluster 11. The genes that we added are *BIRC5*, *KRAS*, *PLK1*, *SS18*, *WDR77*, and *TGIF1*. We have also placed more emphasis on this additional characterization in the manuscript text (page 8, lines 1-4):

“However, we found that cells in cluster 11 were characterized not only by cell cycle genes, but also by the expression of genes involved in chromatin remodeling and organization (CTCF, LAMINB, ATAD2, SS18), regulation of cell proliferation and stemness (HES6 and its target PLK1²⁷, KRAS, FOXM1 and its targets BIRC5, AURKA, AURKB, and CCNA2²⁸), and DNA repair (BRCA2, FANCI, RAD51C, POLQ)(Fig. 4b).”

Figure 6. Gene expression dot plot of a core stem cell-like gene set module based on human and mouse data from Wong et al. [PMID: 18397753] in the scRNA-seq clusters.

Reviewer 3 comment:

In 4F/G authors show RNA velocity and describe it as differentiation trajectory, based on this analysis the differentiation trajectories seem to be different between samples and for example clusters 8/9 in RES-A seem to be less differentiated compared to cells in the same sample in cluster 11 (stem-like). How do the authors reconcile this?

Response: We thank the reviewer for the feedback on Fig. 4f/g and the associated results. The reviewer’s question helped us realize that the UMAP visualizations in Fig. 4f can be misleading due to cells being plotted on top of each other. To better illustrate the results of the analysis, we have revised Fig. 4f to be shown as boxplots, where the CytoTRACE score of each cell in each cluster is plotted and clusters are ordered from least to most differentiated in each sample. We additionally indicate significant differences between cluster mean differentiation values for the least differentiated clusters. As CytoTRACE aims to estimate differentiation potential and is based on the number of detectably expressed genes per cell, we have further modified the discussion of the CytoTRACE results in the text to clarify that the scores generated by the tool are an estimate of developmental potential (page 8, lines 18-21):

“Using CytoTRACE estimation of differentiation states based on the number of expressed genes, cells in cluster 11 showed high developmental potential in most sample conditions (Fig. 4f), suggesting that other cell subpopulations could derive from cells in this cluster.”

We additionally changed the color scheme in Fig. 4g to make the RNA velocity streamlines more visible to the reader.

The analysis in Fig. 4f-g shows that cluster 11 is consistently one of the least differentiated clusters. Cluster 11 is the least differentiated in parental LNCaP, while in LNCaP-RES cells it is the least differentiated together with cluster 9. We acknowledge that cluster 9 is the least differentiated in LNCaP-ENZ48, followed by cluster 11. This might be a reflection of the short term effect of ENZ on the transcriptional configuration of the clusters.

Reviewers' Comments:

Reviewer #1:

Remarks to the Author:

The revised manuscript entitled "Single-cell ATAC and RNA sequencing reveal pre-existing and persistent subpopulations of cells associated with relapse of prostate cancer" provides compelling genomic data on the identification of the genes driving pre-existing (progenitor/) cell populations functionally contributing to the emergence of prostate tumor relapse.

The authors addressed all the issues raised by the three reviewers and the work is found to be highly innovative and of major translational and mechanistic significance in understanding therapeutic resistance and emergence of lethal disease in advanced prostate cancer. The gene profiling and the functional correlations at the single-cell level are beautifully with tumor relapse/resistance outcomes as endpoints.

Reviewer #2:

Remarks to the Author:

the authors have addressed my short comments. Outstanding study, well done to all.

Doug Strand

Reviewer #3:

Remarks to the Author:

The authors have properly addressed my comments.

NCOMMS-21-12818: Point-by-point response to the reviewers' comments

Reviewer #1 (Remarks to the Author):

The revised manuscript entitled "Single-cell ATAC and RNA sequencing reveal pre-existing and persistent subpopulations of cells associated with relapse of prostate cancer" provides compelling genomic data on the identification of the genes driving pre-existing (progenitor/) cell populations functionally contributing to the emergence of prostate tumor relapse.

The authors addressed all the issues raised by the three reviewers and the work is found to be highly innovative and of major translational and mechanistic significance in understanding therapeutic resistance and emergence of lethal disease in advanced prostate cancer. The gene profiling and the functional correlations at the single-cell level are beautifully with tumor relapse/resistance outcomes as endpoints.

Response: We thank the reviewer for their assessment of our manuscript.

Reviewer #2 (Remarks to the Author):

the authors have addressed my short comments. Outstanding study, well done to all.

Doug Strand

Response: We thank the reviewer for his assessment of our manuscript.

Reviewer #3 (Remarks to the Author):

The authors have properly addressed my comments.

Response: We thank the reviewer for their assessment of our manuscript.